# NORM: KNOWLEDGE DISTILLATION VIA N-TO-ONE REPRESENTATION MATCHING

**Xiaolong Liu\*, Lujun Li\*, Chao Li\*, Anbang Yao\*†**
Intel Labs China
{xiaolong.liu,lujun.li,chao3.li,anbang.yao}@intel.com

## ABSTRACT

Existing feature distillation methods commonly adopt the *One-to-one Representation Matching* between any pre-selected teacher-student layer pair. In this paper, we present *N-to-One Representation Matching* (NORM), a new two-stage knowledge distillation method, which relies on a simple Feature Transform (FT) module consisting of two linear layers. In view of preserving the intact information learnt by the teacher network, during training, our FT module is merely inserted after the last convolutional layer of the student network. The first linear layer projects the student representation to a feature space having $N$ times feature channels than the teacher representation from the last convolutional layer, and the second linear layer contracts the expanded output back to the original feature space. By sequentially splitting the expanded student representation into $N$ non-overlapping feature segments having the same number of feature channels as the teacher's, they can be readily forced to approximate the intact teacher representation simultaneously, formulating a novel many-to-one representation matching mechanism conditioned on a single teacher-student layer pair. After training, such an FT module will be naturally merged into the subsequent fully connected layer thanks to its linear property, introducing no extra parameters or architectural modifications to the student network at inference. Extensive experiments on different visual recognition benchmarks demonstrate the leading performance of our method. For instance, the ResNet18|MobileNet|ResNet50-1/4 model trained by NORM reaches 72.14%|74.26%|68.03% top-1 accuracy on the ImageNet dataset when using a pre-trained ResNet34|ResNet50|ResNet50 model as the teacher, achieving an absolute improvement of 2.01%|4.63%|3.03% against the individually trained counterpart. Code is available at https://github.com/OSVAI/NORM.

## 1 INTRODUCTION

Knowledge distillation (KD), an effective way to train compact yet accurate neural networks through knowledge transfer, has attracted increasing research attention recently. Bucilă et al. (2006) and Ba & Caruana (2014) made early attempts in this direction. Hinton et al. (2015) presented the well-known KD using a teacher-student framework, which starts with pre-training a large network (teacher), and then trains a smaller target network (student) on the same dataset by forcing it to match the logits predicted by the teacher model. Many subsequent methods follow this two-stage KD scheme but use hidden layer features as extra knowledge, while others use a one-stage KD scheme in which teacher and student networks are trained from scratch jointly (Guo et al., 2021). In this paper, we focus on two-stage feature distillation (FD) research, mainly for supervised image classification tasks.

Existing two-stage FD methods primarily use feature maps (Romero et al., 2015), or attention maps (Zagoruyko & Komodakis, 2017), or other forms of features (Chen et al., 2021a) at one or multiple hidden layers as knowledge representations. Generally, modern neural network architectures engineered on the ImageNet classification dataset (Russakovsky et al., 2015) adopt a multi-stage design paradigm. At a pair of the same staged layers, a teacher network typically has more output feature channels than a student network while keeping the same spatial feature size. All feature

---

\* XL, LL and CL contributed to the basic method implementations. XL conducted the main experiments. AY proposed the original idea, supervised the project and led the paper writing. † Corresponding author.

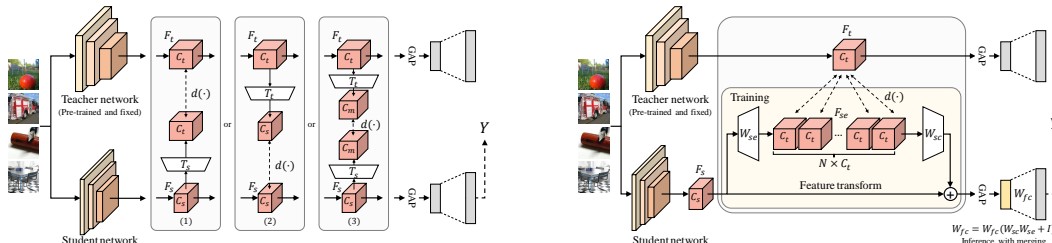

Figure 1: An architectural comparison of prevailing *One-to-one Representation Matching* (ORM) schemes (left figure) and our ***N-to-One Representation Matching*** (NORM, right figure). Based on ORM, existing feature distillation methods apply their feature transforms (FTs) to (1) the student network, or (2) the teacher network, or (3) both of them, at any pre-selected hidden layer pair. Unlike them, NORM leverages a simple linear FT module added after the last convolutional layer of the student network to formulate a many-to-one representation matching scheme via feature expansion, splitting and mimicking. For inference, our FT module will be merged into its subsequent fully connected layer, introducing no extra parameters or architectural modifications to the student network. *Best viewed with zoom in, and see the Method section for a detailed formulation of NORM.*

dimensions may be different at a cross-stage layer pair. To align the feature dimensions, there have been many feature transform (FT) designs. However, prevailing teacher FT designs cause information loss due to dimension reduction, as studied in (Heo et al., 2019a; Tian et al., 2020). More importantly, we observe that existing FD methods adopt the *One-to-one Representation Matching* (ORM) between each pre-selected teacher-student layer pair, indicating that only one knowledge transfer route is introduced. We argue that this leaves considerable room to promote two-stage FD research.

Driven by the above analysis, in this paper, we present a new two-stage feature distillation method dubbed ***N-to-One Representation Matching*** (NORM) that relies on a simple FT module consisting of two linear layers. An architectural comparison of popular ORM schemes and NORM is depicted in Figure 1. When formulating NORM, we leverage three basic principles: (1) using as few FTs as possible; (2) enabling many-to-one feature mimicking flow via student representation expansion and splitting; (3) making FT module absorbable. With the first principle, our FT module is merely inserted after the last convolutional layer of the student network. In this way, the intact information learnt by the teacher network is preserved, and knowledge transfer flow only needs to be considered between the last convolutional layer pair. With the second principle, our FT module starts with a linear layer that projects the student representation to a feature space having $N$ times feature channels than the teacher representation. This allows NORM to introduce many parallel knowledge transfer routes between a single teacher-student layer pair via simple student feature splitting and group-wise feature mimicking operations. With the third principle, our FT module ends with another linear layer that projects the expanded student representation back to the original feature space, and it does not contain any non-linear activation functions, making all of its operations linear. As a result, after training the FT module can be directly merged into its subsequent fully connected layer, without introducing any extra parameters or architectural modifications to the student network at inference.

We evaluate the performance of NORM on different visual recognition benchmarks. On the CIFAR-100 dataset, the student models trained by NORM show a mean accuracy improvement of $2.88\%$ over 7 teacher-student pairs of the same type network architectures. Over 6 teacher-student pairs of different type network architectures, the mean accuracy improvement reaches $5.81\%$, and the maximal gain is $6.92\%$. Leading results are obtained on the large-scale ImageNet dataset. With NORM, the ResNet18|MobileNet|ResNet50-1/4 model reaches $72.14\%|74.26\%|68.03\%$ top-1 accuracy when using a pre-trained ResNet34|ResNet50|ResNet50 model as the teacher, showing $2.01\%|4.63\%|3.03\%$ absolute gain to the baseline model. Thanks to its simplicity and compatibility, we show that improved performance could be further attained by combining NORM with other popular distillation strategies like logits based supervision (Hinton et al., 2015) and contrastive learning (Tian et al., 2020).

## 2    RELATED WORK

**Two-stage KD methods.** This category of KD methods first assumes that a pre-trained teacher network is available, and then uses its learnt representation as extra supervision to guide the training

of a student network. The vanilla KD (Hinton et al., 2015) uses the logits output from the teacher network as soft supervision. FitNets (Romero et al., 2015) show that the feature maps from hidden layers can also be used as hints to improve distillation performance. AT (Zagoruyko & Komodakis, 2017) uses spatial attention maps instead of source feature maps. Many follow-up methods intend to improve the knowledge representation via different techniques, such as feature encoding (Yim et al., 2017; Chen et al., 2021a; Srinivas & Fleuret, 2018), feature selection (Heo et al., 2019a;b; Chen et al., 2021b), distribution learning (Ahn et al., 2019; Huang & Wang, 2017; Passalis & Tefas, 2018; Malinin et al., 2020; Yang et al., 2021), attention rephrasing (Kim et al., 2018; Ji et al., 2021), and reuse of teacher classifier (Chen et al., 2022). Besides, some works (Peng et al., 2019; Yin et al., 2020; Park et al., 2019; Tung & Mori, 2019) explore the use of sample relations. It is also worth noting that several recent works (Yue et al., 2020; Liu et al., 2020b; Deng et al., 2022) propose reinforcement learning based searching methods to improve feature distillation process.

**One-stage KD methods.** This category of KD methods adopts an online training framework that does not require to pre-train the teacher model. ONE (Lan et al., 2018) presents a multi-branch training strategy using an on-the-fly branch ensemble to guide the training of individual branches. DML (Zhang et al., 2018b) uses a mutual learning strategy to jointly train a set of peer models from scratch. During training, every peer model acts as a teacher of the others. DCM (Yao & Sun, 2020) shows that the logits from auxiliary branches of teacher and student networks can improve mutual distillation performance. Many one-stage KD variants have been presented recently, including but not limited to (Guo et al., 2020a; Chung et al., 2020; Malinin et al., 2020; Wu & Gong, 2021).

**Other KD variants.** Extending knowledge distillation methodology from standard supervised learning to other learning scenarios has gained broad attention. BANs (Furlanello et al., 2018) improve the training of a target network via a progressive self-distillation formulation. CRD (Tian et al., 2020) formulates an effective structural representation matching loss based on contrastive learning. Xu et al. (2020) further presented a way to combine self-supervision and contrastive learning. Besides, there also exist various KD variants for life-long learning (Li & Hoiem, 2016), semi-supervised learning (Pham et al., 2021), few-shot learning (Shen et al., 2021), data-free learning (Lopes et al., 2017), adversarial learning (Chung et al., 2020) and distributed learning (Anil et al., 2018). Although these methods are still tailored to image classification, KD has also been applied to handle other tasks, such as semantic segmentation (Liu et al., 2019b), object detection (Wang et al., 2019), image super-resolution (Liu et al., 2020a), and neural machine translation (Kim & Rush, 2016).

Our work focuses on two-stage KD research, mainly for supervised image classification tasks. Specifically, we attempt to shift the prevailing one-to-one feature representation matching paradigm to a many-to-one alternative conditioned on a single teacher-student layer pair. This makes our method differ with existing KD methods both in motivation and formulation.

## 3 METHOD

### 3.1 BACKGROUND: ONE-TO-ONE REPRESENTATION MATCHING

We first review the representation matching of existing two-stage feature distillation (FD) methods in a general formulation. Suppose that we have a pre-trained teacher network $T$, a target student network $S$, a given image classification dataset $X$ and its ground truth label set $Y$. For a pre-selected teacher-student layer pair (two same staged layers of $T$ and $S$, as default), let $F_t \in \mathbb{R}^{H \times W \times C_t} | F_s \in \mathbb{R}^{H \times W \times C_s}$ denote the output feature maps for the teacher|student network, where $H$, $W$ and $C_t | C_s$ are the channel height, width and number, respectively. Let $T_t | T_s$ denote the feature transform (FT) for the teacher|student network, which projects $F_t | F_s$ to the same feature space, respectively. Then, the representation matching loss of existing FD methods, in general, can be defined as:

$$L_{fd} = d(T_s(F_s), T_t(F_t)). \quad (1)$$

Various distance metrics $d(\cdot)$, such as $l_2$-norm distance (Romero et al., 2015; Zagoruyko & Komodakis, 2017; Heo et al., 2019b), $l_1$-norm distance (Kim et al., 2018) and maximum mean discrepancy (Huang & Wang, 2017) are popularly used in FD research. Furthermore, existing FD methods usually apply the feature representation matching to one or multiple pre-selected teacher-student layer pairs. Under this context, it is clear that the use of FT and the design of FT are two key problems for the feature representation matching. To the first problem, there are three choices: applying one or multiple FTs to (1) the student network (Romero et al., 2015); (2) the teacher network (Yue et al.,

2020); (3) both teacher and student networks (Zagoruyko & Komodakis, 2017; Heo et al., 2019b; Srinivas & Fleuret, 2018; Heo et al., 2019a; Chen et al., 2021a). They are illustrated in Figure 1 (left figure). Comparatively, the last choice is much more popular than the other two. To the second problem, there exist many FT designs (Guo et al., 2021) that generate aligned feature maps, or attention maps, or other forms of features used as the knowledge representation for matching. However, according to Eq. 1, we can see that existing FD methods use the *One-to-one Representation Matching* (ORM). More specifically, they perform global single-shot feature mimicking process, meaning that there exists only one knowledge transfer route between any teacher-student layer pair. We conjecture that if we can formulate an effective mechanism to introduce multiple global knowledge transfer routes between any teacher-student layer pair, it will offer more chances to inject the intact teacher's knowledge into the student network, and improved FD performance could be attained.

## 3.2 N-TO-ONE REPRESENTATION MATCHING

Motivated by the above analysis, we present ***N-to-One Representation Matching*** (NORM), a new two-stage feature distillation method that relies on a simple FT module. In the formulation of NORM, we rethink the representation matching mechanism from the following perspectives: (1) regarding the use of FT, where to place it? (2) regarding the design of FT, how to adapt it for introducing many parallel global knowledge transfer routes between any pre-selected teacher-student layer pair? (3) regarding the distillation performance, how to make the accuracy-efficiency tradeoff? To the first and third questions, our basic principle is *"using as few FTs as possible"*, partially inspired by (Romero et al., 2015; Heo et al., 2019a; Tian et al., 2020). Accordingly, we merely insert an FT module after the last convolutional layer of the student network. In this way, NORM preserves the intact information learnt by the pre-trained teacher network, and knowledge transfer flow only needs to be considered between the last convolutional layer pair. Note that the design of directly plugging an FT module into the student network is in sharp contrast to existing FD methods which typically use their FTs as auxiliary branches that will be discarded at inference. To the second question, our basic principle is *"enabling many-to-one knowledge mimicking flow via student representation expansion and splitting"*. Accordingly, we construct a student FT module which starts with a linear layer that projects the student representation to a feature space having $N$ times feature channels than the teacher representation. By sequentially splitting the expanded student representation into $N$ non-overlapping segments having the same number of feature channels as the teacher's, we can force them to approximate the intact teacher representation simultaneously, formulating an effective many-to-one feature matching mechanism conditioned on a single teacher-student layer pair. To the third question, we additionally use a *"making FT module absorbable"* principle. Accordingly, our student FT module ends with another linear layer that projects the expanded student representation back to the original feature space, and it does not contain any non-linear activation functions. As a result, such an FT module will be naturally merged into its subsequent fully connected layer after training, without introducing any extra parameters or architectural modifications to the student network at inference. An overview of NORM is depicted in Figure 1 (right figure).

Now, we provide the formulation of NORM following the notations in Eq. 1. Given a teacher-student layer pair (last convolutional layer pair, as default) with the output feature maps $F_t \in \mathbb{R}^{H \times W \times C_t}$ and $F_s \in \mathbb{R}^{H \times W \times C_s}$, let $T_s(W_{se}, W_{sc})$ denote our student FT module consisting of two linear layers. The first linear layer uses a $1 \times 1$ convolutional kernel $W_{se} \in \mathbb{R}^{1 \times 1 \times C_s \times NC_t}$ along the channel dimension to project each pixel in $F_s$ to a desired channel dimension $NC_t$, producing an expanded student representation $F_{se} \in \mathbb{R}^{H \times W \times NC_t}$ having $N$ times feature channels than the teacher representation. The second linear layer uses another $1 \times 1$ convolutional kernel $W_{sc} \in \mathbb{R}^{1 \times 1 \times NC_t \times C_s}$ to project each pixel in $F_{se}$ back to the original channel dimension $C_s$, producing $F_{sc} \in \mathbb{R}^{H \times W \times C_s}$. Mathematically, the student FT module with the input $F_s$ can be written as:

$$F_{se} = W_{se} * F_s, \quad F_{sc} = W_{sc} * F_{se}, \tag{2}$$

where $*$ denotes the convolution operation. Next, we sequentially split the expanded student representation $F_{se}$ into $N$ non-overlapping segments $F_{se}^i \in \mathbb{R}^{H \times W \times C_t}, 1 \le i \le N$ having the same number of feature channels as the teacher's. This allows us to force $N$ student feature segments to approximate the intact teacher representation simultaneously by minimizing an $l_2$-norm distance metric, formulating our many-to-one representation matching mechanism conditioned on a single

teacher-student layer pair. Specifically, our many-to-one representation matching loss is defined as:

$$L_{norm} = \frac{1}{N} \sum_{i=1}^{N} ||F_{se}^i - F_t||_2^2, \tag{3}$$

and the total training loss of NORM to be minimized is defined as:

$$L_{total} = L_{ce} + \alpha L_{norm}, \tag{4}$$

where $L_{ce}$ denotes the standard cross-entropy loss of the student network supervised by the ground truth labels, and $\alpha$ is a positive coefficient ($\alpha = 10$, as default) to weight the loss $L_{norm}$.

**Interpretation of NORM.** The formulation of NORM can be interpreted as a novel way of learning a dynamically mixed feature ensemble over multiple augmented views of the same student representation by forcing them to mimic the intact teacher representation simultaneously. More precisely, in NORM, the first linear layer $W_{se}$ acts as a set of independently initialized feature transforms to generate $N$ channel-expanded views of the student feature $F_s$ whose representation abilities are then parallelly augmented by the distillation supervision from the intact teacher feature $F_t$, and the second linear layer $W_{sc}$ performs a learnable ensemble of $N$ distillation-augmented student feature views via fully connected channel mixing operations (which make the feature ensemble $F_{sc}$ has the same size to $F_s$, guaranteeing the absorbable property of NORM at inference). Our mixed student feature ensemble learning further benefits from the standard cross-entropy loss supervised by the ground truth labels. In order to understand what enables the distillation effectiveness, we systematically study the major components of NORM in the Experiments section.

**Augmented NORM.** Thanks to its simplicity, NORM can be easily augmented by additionally introducing a vanilla logits based KD loss (at the network head) (Hinton et al., 2015) into Eq. 4:

$$L_{total} = L_{ce} + \alpha L_{norm} + \beta L_{kd}, \tag{5}$$

where $L_{kd}$ matches the logits distribution of the student network to a target distribution produced by the teacher network, and $\beta$ is a positive coefficient ($\beta = 4$, as default). Besides, NORM can be also combined with a contrastive KD loss (Tian et al., 2020) for improved results, as tested in experiments.

**Implementation and Inference.** As our FT module does not have any non-linear activation functions, we empirically find that inserting it after the last convolutional layer of different student networks will mostly lead to obvious accuracy drop in the standard training regime (individually train the student model), but with NORM the performance of final student models will be improved significantly. To suppress the accuracy drop issue, we add a linear residual connection (identity mapping, as shown in Figure 1) from the input to the output of our student FT module, while maintaining the absorbable property. In implementation, we use this student FT module as our default setting to NORM, and set $N = 8$ for all main experiments (the choice of $N$ is studied in Figure 2). For inference, let $W_{fc} \in \mathbb{R}^{C_s \times C_{fc}}$ denote the learnt parameters of the fully connected (FC) layer after the student FT module $T_s(W_{se}, W_{sc})$, we can directly merge the student FT module into its subsequent FC layer by $W_{fc} = W_{fc}(W_{sc}W_{se} + I)$, where $I \in \mathbb{R}^{C_s \times C_s}$ is an identity matrix. Note that modern neural networks typically have a global average pooling layer before the FC layer, and it does not affect the absorbable property of our student FT module at inference. *We put the proofs of them in the Appendix.*

**Differences with network re-parameterization methods.** The absorbable property of our student FT module is based on the network re-parameterization. In recent years, there have been lots of network re-parameterization methods that are presented in the deep learning field. ACNet (Ding et al., 2019) uses an absorbable multi-branch block based on 1D asymmetric convolutions to replace the square convolution. RepVGG (Ding et al., 2021) presents an absorbable multi-branch block to replace a stack of $3 \times 3$ convolutions in VGG-like networks (Simonyan & Zisserman, 2015). This category of methods focuses on strengthening the learning power of basic convolutions via equivalent multi-branch alternatives, particularly from the perspective of network structure engineering in the standard training regime. Clearly, our method differs with them both in focus, formulation and application. ExpandNets (Guo et al., 2020b) and WIN (Zhou et al., 2020) leverage over-parameterization designs based on linear expansion and contraction to improve the training of a thin network, which are more closely related to our work. ExpandNets expand all convolutional and FC layers during training, and convert the expanded network back to the original one at inference. WIN first trains a wider network generated by uniformly expanding the width of all building blocks in a given thin network, and uses it as the teacher. Then, it inserts a pair of linear expansion and contraction layers between

Table 1: Top-1 mean accuracy (%) comparison on CIFAR-100. The teacher and student have the same type network architectures. The results of the current mainstream KD methods are obtained from the papers of CRD, SemCKD, ReviewKD, SimKD and DistPro. The plain FT and the default FT denote our student feature transform module without and with a linear residual connection, respectively. NORM+KD and NORM+CRD denote combining NORM with the vanilla logits based KD and the contrastive KD, respectively. The best and second best results are bolded and underlined, respectively.

| Teacher
Student | WRN-40-2
WRN-16-2 | WRN-40-2
WRN-40-1 | ResNet56
ResNet20 | ResNet110
ResNet20 | ResNet110
ResNet32 | ResNet32x4
ResNet8x4 | VGG13
VGG8 |
|---|---|---|---|---|---|---|---|
| Teacher | 75.61 | 75.61 | 72.34 | 74.31 | 74.31 | 79.42 | 74.64 |
| Student (reported in CRD) | 73.26 | 71.98 | 69.06 | 69.06 | 71.14 | 72.50 | 70.36 |
| Student (our reproduced) | 73.80 | 71.70 | 69.53 | 69.53 | 71.56 | 72.87 | 70.75 |
| Student (w/ 1 plain FT) | 72.59 | 71.14 | 68.09 | 68.09 | 70.17 | 73.51 | 70.17 |
| Student (w/ 1 default FT) | 73.72 | 72.09 | 69.55 | 69.55 | 71.64 | 73.72 | 70.64 |
| KD | 74.92 | 73.54 | 70.66 | 70.67 | 73.08 | 73.33 | 72.98 |
| FitNet | 73.58 | 72.24 | 69.21 | 68.99 | 71.06 | 73.50 | 71.02 |
| AT | 74.08 | 72.77 | 70.55 | 70.22 | 72.31 | 73.44 | 71.43 |
| SP | 73.83 | 72.43 | 69.67 | 70.04 | 72.69 | 72.94 | 72.68 |
| CC | 73.56 | 72.21 | 69.63 | 69.48 | 71.48 | 72.97 | 70.71 |
| VID | 74.11 | 73.30 | 70.38 | 70.16 | 72.61 | 73.09 | 71.23 |
| RKD | 73.35 | 72.22 | 69.61 | 69.25 | 71.82 | 71.90 | 71.48 |
| PKT | 74.54 | 73.45 | 70.34 | 70.25 | 72.61 | 73.64 | 72.88 |
| AB | 72.50 | 72.38 | 69.47 | 69.53 | 70.98 | 73.17 | 70.94 |
| FT | 73.25 | 71.59 | 69.84 | 70.22 | 72.37 | 72.86 | 70.58 |
| FSP | 72.91 | n/a | 69.95 | 70.11 | 71.89 | 72.62 | 70.23 |
| NST | 73.68 | 72.24 | 69.60 | 69.53 | 71.96 | 73.30 | 71.53 |
| CRD | 75.48 | 74.14 | 71.16 | 71.46 | 73.48 | 75.51 | 73.94 |
| SRRL | n/a | 74.64 | n/a | n/a | n/a | 75.39 | n/a |
| SemCKD | n/a | 74.41 | n/a | n/a | n/a | 76.23 | 74.43 |
| ReviewKD | 76.12 | 75.09 | 71.89 | n/a | 73.89 | 75.63 | **74.84** |
| SimKD | n/a | **75.56** | n/a | n/a | n/a | **78.08** | n/a |
| DistPro | **76.36** | n/a | **72.03** | n/a | 73.74 | n/a | n/a |
| NORM (w/ 1 plain FT) | 75.57 | 74.78 | 70.70 | 71.01 | 73.27 | 76.76 | 73.64 |
| NORM | 75.65 | 74.82 | 71.35 | 71.55 | 73.67 | 76.49 | 73.95 |
| NORM+KD | 76.26 | 75.42 | 71.61 | **72.00** | **73.95** | 76.98 | 74.46 |
| NORM+CRD | 76.02 | 75.37 | 71.51 | 71.90 | 73.81 | 76.49 | 73.58 |

any two neighboring building blocks of the given thin network, and considers the sequential stacks of expanded building blocks as a set of subnetworks, which are trained progressively one by one using the one-to-one feature representation matching. ExpandNets and WIN all adopt the vanilla KD (Hinton et al., 2015) to further fine-tune their trained models. In sharp contrast to them, our method aims to advance two-stage feature distillation research by presenting a novel many-to-one representation matching strategy conditioned on a single teacher-student layer pair. Accordingly, our method inserts only one linear FT module to the last convolutional layer of a student network, without need of multi-layer feature mimicking and progressive training with multiple restarts. Furthermore, our method is applicable to various teacher-student pairs with both the same type and different type network architectures. Comparatively, our method is a more simple and easy to use, and it (without the vanilla KD) outperforms ExpandNets and WIN with large margins on ImageNet (see Table 6).

## 4 EXPERIMENTS

### 4.1 PERFORMANCE ON IMAGE CLASSIFICATION TASK

**Datasets and experimental setups.** We use CIFAR-100 (Krizhevsky & Hinton, 2009) and ImageNet (Russakovsky et al., 2015) datasets for basic experiments. CIFAR-100, which consists of 50,000 training images and 10,000 test images with 100 classes, is a popular classification dataset for KD research. Following the settings of CRD (Tian et al., 2020), we use 13 teacher-student pairs having either the same type or different type network architectures (see Table 1,2) for experiments. Each experiment is conducted for 5 separate runs, and we report top-1 mean recognition rate on the test set. ImageNet contains over 1.2 million images for training and 50,000 images for validation, including 1,000 image classes. Comparatively, ImageNet is much more challenging than CIFAR-100. Following the settings of Tian et al. (2020); Yang et al. (2021); Chen et al. (2021b), we use 2 popular teacher-student pairs (see Table 3) for experiments. We report top-1 recognition rate on the validation set. For comprehensive comparisons, we compare our method with the current mainstream two-stage and one-stage KD methods, including the vanilla KD (Hinton et al., 2015), FitNets (Romero et al., 2015), AT (Zagoruyko & Komodakis, 2017), SP (Tung & Mori, 2019), CC (Peng et al., 2019), VID (Ahn et al., 2019), RKD (Park et al., 2019), PKT (Passalis & Tefas, 2018), AB (Heo et al., 2019b), FT (Kim et al., 2018), FSP (Yim et al., 2017), NST (Huang & Wang, 2017), CRD (Tian et al., 2020), OFD (Heo et al., 2019a), SSKD (Xu et al., 2020), ONE (Lan et al., 2018), PCL (Wu & Gong, 2021), SRRL (Yang et al., 2021), ReviewKD (Chen et al., 2021b), SemCKD (Chen et al., 2021a),

Table 2: Top-1 mean accuracy (%) comparison on CIFAR-100. The teacher and student have different type network architectures. Basic settings are the same to those described in the caption of Table 1.

| Teacher
Student | VGG13
MobileNetV2 | ResNet50
MobileNetV2 | ResNet50
VGG8 | ResNet32x4
ShuffleNetV1 | ResNet32x4
ShuffleNetV2 | WRN-40-2
ShuffleNetV1 |
|---|---|---|---|---|---|---|
| Teacher | 74.64 | 79.34 | 79.34 | 79.42 | 79.42 | 75.61 |
| Student (reported in CRD) | 64.60 | 64.60 | 70.36 | 70.50 | 71.82 | 70.50 |
| Student (our reproduced) | 64.81 | 64.81 | 70.75 | 71.63 | 72.96 | 71.63 |
| Student (w/ 1 plain FT) | 63.95 | 63.95 | 70.17 | 71.82 | 72.55 | 71.82 |
| Student (w/ 1 default FT) | 64.13 | 64.13 | 70.64 | 71.76 | 72.71 | 71.76 |
| KD | 67.37 | 67.35 | 73.81 | 74.07 | 74.45 | 74.83 |
| FitNet | 64.14 | 63.16 | 70.69 | 73.59 | 73.54 | 73.73 |
| AT | 59.40 | 58.58 | 71.84 | 71.73 | 72.73 | 73.32 |
| SP | 66.30 | 68.08 | 73.34 | 73.48 | 74.56 | 74.52 |
| CC | 64.86 | 65.43 | 70.25 | 71.14 | 71.29 | 71.38 |
| VID | 65.56 | 67.57 | 70.30 | 73.38 | 73.40 | 73.61 |
| RKD | 64.52 | 64.43 | 71.50 | 72.28 | 73.21 | 72.21 |
| PKT | 67.13 | 66.52 | 73.01 | 74.10 | 74.69 | 73.89 |
| AB | 66.06 | 67.2 | 70.65 | 73.55 | 74.31 | 73.34 |
| FT | 61.78 | 60.99 | 70.29 | 71.75 | 72.50 | 72.03 |
| NST | 58.16 | 64.96 | 71.28 | 74.12 | 74.68 | 74.89 |
| CRD | 69.73 | 69.11 | 74.30 | 75.11 | 75.65 | 76.05 |
| SRRL | n/a | n/a | n/a | 75.18 | n/a | n/a |
| SemCKD | n/a | n/a | n/a | n/a | 77.62 | n/a |
| ReviewKD | **70.37** | 69.89 | n/a | 77.45 | 77.78 | 77.14 |
| SimKD | n/a | n/a | n/a | 77.18 | n/a | n/a |
| DistPro | n/a | n/a | n/a | 77.18 | 77.54 | 77.24 |
| NORM (w/ plain FT) | 69.37 | 70.94 | 74.37 | 75.93 | 77.34 | 76.61 |
| NORM | 68.94 | 70.56 | 75.17 | 77.42 | 78.07 | 77.06 |
| NORM+KD | 69.38 | **71.17** | **75.67** | **77.79** | **78.32** | **77.63** |
| NORM+CRD | 69.17 | 71.08 | 75.51 | 77.50 | 77.96 | 77.09 |

Table 3: Top-1 accuracy (%) comparison on ImageNet. The results in the bracket are for our reproduced student baselines, and the results of the current mainstream KD methods are obtained from the papers of CRD, SSKD, SRRL, SemCKD, ReviewKD, SimKD and DistPro.

| Teacher | Student | CC | SP | ONE | PCL | SSKD | KD | AT | OFD | RKD | CRD | SRRL | SemCKD | ReviewKD | SimKD | DistPro | NORM |
|---|---|---|---|---|---|---|---|---|---|---|---|---|---|---|---|---|---|
| ResNet34 \| 73.31 | ResNet18 \| (70.13) | 69.96 | 70.62 | 70.55 | 70.42 | 71.62 | 70.68 | 70.59 | 71.08 | 71.34 | 71.17 | 71.73 | 70.87 | 71.61 | 71.66 | 71.89 | **72.14** |
| ResNet50 \| 76.16 | MobileNet \| (69.63) | n/a | n/a | n/a | n/a | n/a | 70.68 | 70.72 | 71.25 | 71.32 | 71.40 | 72.49 | n/a | 72.56 | n/a | 73.26 | **74.26** |

SimKD (Chen et al., 2022) and DistPro (Deng et al., 2022). All experiments are implemented with PyTorch (Paszke et al., 2019). *Experimental details are put in the Appendix.*

**Results on CIFAR-100.** Table 1 shows the results comparison on CIFAR-100 with 7 teacher-student pairs having the same type network architectures. In average, NORM brings 2.88% top-1 gain to the baseline student models, with the maximal gain of 3.99%. Table 2 provides the results comparison with 6 teacher-student pairs having different type network architectures. In average, NORM brings 5.81% top-1 gain to the baseline student models, with the maximal gain of 6.92%. Generally, NORM shows very competitive results compared to the current mainstream KD methods which usually adopt multi-layer feature distillation schemes. Note that many top KD methods use the vanilla logits based distillation (Hinton et al., 2015) as the extra supervision. Besides, CRD, SRRL and SSKD utilize the contrastive learning to augment knowledge distillation process. In Table 1 and Table 2, we also explore the compatibility of NORM with these two types of distillation regularization (denoted as NORM+KD and NORM+CRD). We can see that: (1) the vanilla logits based distillation can further improve the performance of NORM, showing 0.46% extra gain in average, with the maximal extra gain of 0.61%; (2) the contrastive learning can also improve the performance of NORM in most cases, showing 0.18% extra gain in average, with the maximal extra gain of 0.55%. Finally, NORM+KD achieves the best and the second best results on 7 and 4 of 13 teacher-student pairs, respectively.

**Results on ImageNet.** Table 3 shows the results comparison on ImageNet. We can see that our method always shows the best performance compared to these competing KD methods. Specifically, for the teacher-student pair of ResNet34→ResNet18, NORM improves top-1 accuracy of the student model from 70.13% to 72.14%, outperforming the current best method by a margin of 0.25%. Taking a pre-trained ResNet50 as the teacher model and a MobileNet (Howard et al., 2017) as the target student, the MobileNet model trained by NORM attains 74.26% top-1 accuracy, showing 4.63% gain to the baseline model trained individually. Compared to ResNet34→ResNet18, the capacity gap of the teacher and student network becomes larger for ResNet50→MobileNet. Under this context, the top-1 margin of NORM against the current best method is pronounced, reaching 1.00%.

## 4.2 ABLATION STUDY

To have a deep analysis of NORM, we further provide a lot of ablative experiments mostly performed on CIFAR-100, unless otherwise stated. For the experiments on CIFAR-100, we run our method 5 times for each setting with random initialized seeds, and report top-1 mean recognition rate.

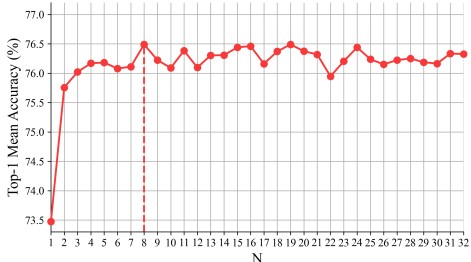

Figure 2: The selection of $N$. On CIFAR-100 with ResNet32x4→ResNet8x4, we compare the performance of NORM with different $N$ settings.

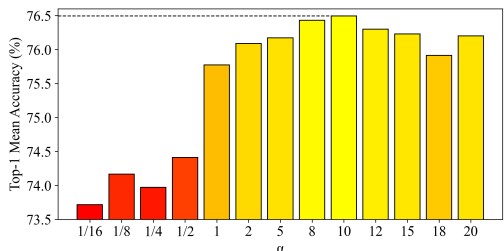

Figure 3: The selection of $\alpha$. On CIFAR-100 with ResNet32x4→ResNet8x4, we compare the performance of NORM ($N = 8$) by changing $\alpha$.

Table 4: Top-1 accuracy (%) comparison of applying NORM to more teacher-student layer pairs. On ImageNet with ResNet34→ResNet18, we insert 2 FT modules ($N = 8$) after conv5_x (our default design) and conv4_x of ResNet18, and compare the following two settings: (1) using NORM to the same staged teacher-student layer pairs; (2) forcing the student representations expanded from conv4_x and conv5_x to match the teacher representation learnt from conv5_x. Plain FT does not have a linear residual connection.

Figure 4: The role of many-to-one representation matching. On CIFAR-100 with ResNet32x4→ResNet8x4, we compare the performance of NORM using $n$ ($1 \leq n \leq N$) of $N = 8$ student feature segments to match the teacher representation.

| Layer pair# | 1 (default) | 2 (same staged) | 2 (cross staged) |
|---|---|---|---|
| NORM (w/ plain FT) | 71.42 | 71.40 | 70.89 |
| NORM | 72.14 | 72.03 | 71.49 |

**The selection of $N$ and $\alpha$.** In our formulation, NORM has two hyper-parameters: $N$ to control the desired number of representation matching routes between a single teacher-student layer pair and $\alpha$ to weight the many-to-one representation matching loss $L_{norm}$. Accordingly, our first two sets of ablative experiments on CIFAR-100 are conducted for the selection of $N$ and $\alpha$. Specifically, we use a teacher-student pair of ResNet32x4→ResNet8x4. Figure 2 and Figure 3 show the performance comparison of NORM with different settings of $N$ and $\alpha$, respectively. From Figure 2, we can observe that: (1) increasing $N$ from 1 to 2 brings $2.28\%$ extra accuracy gain; (2) this gain becomes larger when $N$ gradually increases, and reaches the first peak value at $N = 8$ ($3.02\%$ to $N = 1$ and $3.62\%$ to the baseline student model). To balance training accuracy and efficiency, we set $N = 8$ for the experiments both on CIFAR-100 and ImageNet. According to Figure 3, we typically set $\alpha = 10$ for the experiments on CIFAR-100. For the experiments on ImageNet, we empirically set $\alpha = 8$.

**The role of the many-to-one representation matching.** In NORM, the many-to-one representation matching is performed after sequentially splitting the expanded student representation into $N$ non-overlapping segments having the same number of feature channels as the teacher's. Given $N$ student segments, is it necessary to force them to simultaneously approximate the teacher representation? Again, we use the teacher-student pair of ResNet32x4→ResNet8x4 with $N = 8$ to explore this question. From Figure 4, we can see: the more the segments used to match the teacher representation, the larger the accuracy gain. Specifically, using all 8 segments outperforms using only 1 of 8 segments by a margin of $0.98\%$. Moreover, using 1 of 8 segments for representation matching ($75.51\%$ vs. $73.47\%$) is much better than $N = 1$ in Figure 2. These results validate the advantage of our design.

**Applying NORM to more teacher-student layer pairs.** Recall that NORM performs the proposed many-to-one representation matching between a single teacher-student layer pair, as our FT module is merely inserted after the last convolutional layer of the student network. A natural question is how about the performance when applying NORM to multiple teacher-student layer pairs. We study this question on ImageNet with ResNet34 as teacher and ResNet18 as student. In the experiments, we add two FT modules ($N = 8$) after conv5_x (our default design) and conv4_x of ResNet18, and consider the following two settings: (1) applying NORM to the same staged teacher-student layer pairs simultaneously; (2) forcing two different staged student representations to simultaneously match the teacher representation from conv5_x. *Architectural illustrations of these two NORM variants are referred to the Appendix*. From the results shown in Table 4 we can see that these two variant

Table 5: The effect of linear FT modules to top-1 baseline accuracy (%). On ImageNet, we insert 1 or 2 our linear FT modules into the ResNet-18 network, and train it from scratch individually. Plain FT does not have a linear residual connection.

| FT module number | 0 (baseline) | 1 | 2 |
|---|---|---|---|
| ResNet18 (w/ plain FT) | 70.13 | 69.20 | 68.70 |
| ResNet18 (w/ default FT) | 70.13 | 70.40 | 70.39 |

Table 6: Top-1 accuracy (%) comparison of NORM with network re-parameterization methods using KD. The results of ExpandNets and Win on ImageNet are obtained from their original papers (Guo et al., 2020b; Zhou et al., 2020).

| Student model | ExpandNets+KD | WIN+KD | NORM |
|---|---|---|---|
| MobileNet | 70.47 | N/A | 74.26 |
| ResNet50-1/4 | N/A | 67.50 | 68.03 |

designs lead to worse results than our default design no matter the FT module has a linear residual connection or not, although both of them improve the student accuracy. At first blush, this might seem surprising, since multi-layer feature matching strategies are widely used in KD research (Zagoruyko & Komodakis, 2017; Yim et al., 2017; Heo et al., 2019b; Srinivas & Fleuret, 2018; Heo et al., 2019a; Chen et al., 2021a). As we explained next, this is due to our linear FT modules.

**The effect of linear FT modules.** Someone may concern that the model accuracy improvement may mainly from inserting a linear FT module into each student network. Accordingly, we conduct ablative experiments on ImageNet to explore this concern. In the experiments, we add 1 or 2 linear FT modules into ResNet18 (after conv4_x and conv5_x) first, then train each of them from scratch individually. We consider the linear FT module without or with a linear residual connection, separately. Table 5 shows the results. We can observe that adding one linear FT module without a linear residual connection into ResNet18 leads to obvious accuracy drop (0.93%), and the accuracy drop becomes more serious when adding two this type FT modules. The accuracy drop issue is suppressed by our default FT module having a linear residual connection which only brings marginal model accuracy improvement. Similarly, adding two default FT modules into ResNet18 shows slightly worse model accuracy compared to just adding one. Furthermore, in Table 1 and Table 2, we provide the individually trained model results when adding our two types of FT modules separately to many different student networks on CIFAR-100, and similar observations can be found. These experimental observations are mainly due to the structure of our FT modules. Note that our FT design does not contain popular non-linear activation functions which are critical to stabilize and improve the training process, in order to gain the absorbable property for maintaining efficient inference. Because of this, applying NORM to more teacher-student layer pairs does not achieve further improved performance compared to NORM conditioned on the last convolutional pair, as shown in Table 4.

**Performance comparison with network re-parameterization methods.** In Section 3.2, we discuss the connections and differences of NORM and network re-parameterization methods. In Table 6, we further compare their performance on ImageNet. Clearly, our method obtains more accurate student models than top network re-parameterization methods that also apply knowledge distillation during training. Specifically, for the MobileNet (student), NORM without KD reaches 74.26% top-1 accuracy with a pre-trained ResNet50 as teacher, while ExpandNets (Guo et al., 2020b) with KD gets 70.47% top-1 accuracy using a pre-trained ResNet152 as teacher; for the ResNet50-1/4 (student), NORM without KD reaches 68.03% top-1 accuracy with a pre-trained ResNet50 (teacher), while WIN (Zhou et al., 2020) with KD gets 67.50% top-1 accuracy using a pre-trained wide teacher which is 4 times larger than the student.

**More experiments and discussions.** *Please note that in the Appendix*, we provide more ablative experiments, for a better understanding of NORM. The limitations of NORM are also discussed.

## 5 CONCLUSION

In this paper, we present NORM, a new two-stage feature distillation method. It relies on a linear feature transform module inserted after the last convolutional layer of the student network, and enables a novel many-to-one representation matching mechanism conditioned on a single teacher-student layer pair via feature expansion, splitting and group-wise mimicking. Thanks to its linear property, after training such a feature transform module will be naturally merged into the subsequent FC layer, maintaining the same student network architecture at inference. Extensive experiments on popular image recognition benchmarks show that NORM can attain promising performance in both distillation accuracy and efficiency. We hope NORM would inspire the community to pay more attention to many-to-one representation matching research.

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

# A  APPENDIX

## A.1  PROOFS OF THE ABSORBABLE PROPERTY OF OUR FEATURE TRANSFORM MODULES

Here we provide the proofs to show that our feature transform (FT) module without or with a linear residual connection (identity mapping) could be merged into the subsequent fully connect (FC) layer of the student network after training.

Given a student network $S$, let $T_s(W_{se}, W_{sc})$ denote an FT module (without a linear residual connection) inserted after the last convolutional layer of $S$, let $W_{fc}$ denote the weight matrix of the subsequent FC layer, let $F \in \mathbb{R}^{H \times W \times C_s}$ denote the input feature maps of $T_s$, where $H$, $W$ and $C_s$ are the channel height, width and number, respectively. For inference, the mathematical operations from the FT module to the subsequent FC layer can be defined as:

$$F^{'} = W_{sc} * (W_{se} * F), \quad O = W_{fc}(GAP(F^{'})), \tag{6}$$

where $*$ denotes the convolution operation, $W_{se} \in \mathbb{R}^{1 \times 1 \times C_s \times NC_t}, W_{sc} \in \mathbb{R}^{1 \times 1 \times NC_t \times C_s}$ denote two point-wise convolutional kernels for channel expansion and contraction, respectively; $W_{fc} \in \mathbb{R}^{C_s \times C_{fc}}$ in the FC layer can be also seen as a point-wise convolutional kernel; $GAP$ denotes global average pooling operation; $F^{'} \in \mathbb{R}^{H \times W \times C_s}$ denotes the output feature maps of the FT module; $O \in \mathbb{R}^{C_{fc}}$ denotes the output vector of the FC layer.

We use $F_{i,j} \in \mathbb{R}^{C_s}$ to denote the vector of channel pixels at spatial location $(i, j)$ of $F$. For the point-wise convolutional kernel $W_{se}$, we have:

$$(W_{se} * F)_{i,j} = W_{se} F_{i,j}. \tag{7}$$

Therefore, the vector of channel pixels at spatial location $(i, j)$ of $F^{'}$ can be calculated as:

$$F^{'}_{i,j} = (W_{sc} * (W_{se} * F))_{i,j} = W_{sc}(W_{se} F_{i,j}) = (W_{sc} W_{se}) F_{i,j}, \tag{8}$$

then we can write:

$$\begin{aligned}
O &= W_{fc} GAP(F^{'}) \\
&= W_{fc} \sum_{i=1}^{H} \sum_{j=1}^{W} \frac{F^{'}_{i,j}}{HW} \\
&= W_{fc} \sum_{i=1}^{H} \sum_{j=1}^{W} \frac{(W_{sc} W_{se}) F_{i,j}}{HW} \\
&= (W_{fc} W_{sc} W_{se}) \sum_{i=1}^{H} \sum_{j=1}^{W} \frac{F_{i,j}}{HW} \\
&= W^{'}_{fc} GAP(F),
\end{aligned} \tag{9}$$

where $W^{'}_{fc} = W_{fc} W_{sc} W_{se}$. Now, it is clear that we can directly merge the FT module (without a linear residual connection) into its subsequent FC layer of the student network at inference.

For the FT module with a linear residual connection, the calculation of $F^{'}$ is defined as:

$$\begin{aligned}
F^{'}_{i,j} &= (W_{sc} * (W_{se} * F) + F)_{i,j} \\
&= (W_{sc}(W_{se} F))_{i,j} + F_{i,j} \\
&= (W_{sc} W_{se}) F_{i,j} + F_{i,j} \\
&= (W_{sc} W_{se} + I) F_{i,j},
\end{aligned} \tag{10}$$

where $I \in \mathbb{R}^{C_s \times C_s}$ is an identity matrix. Then we have:

$$
\begin{aligned}
O &= W_{fc} \sum_{i=1}^{H} \sum_{j=1}^{W} \frac{F'_{i,j}}{HW} \\
&= W_{fc} \sum_{i=1}^{H} \sum_{j=1}^{W} \frac{(W_{sc}W_{se} + I)F_{i,j}}{HW} \\
&= W_{fc}(W_{sc}W_{se} + I) \sum_{i=1}^{W} \sum_{j=1}^{W} \frac{F_{i,j}}{HW} \\
&= W'_{fc} GAP(F).
\end{aligned}
\tag{11}
$$

where $W'_{fc} = W_{fc}(W_{sc}W_{se} + I)$. Again, it is clear that we can directly merge the FT module (with a linear residual connection) into its subsequent FC layer of the student network at inference.

## A.2   DATASETS AND IMPLEMENTATION DETAILS

Recall that our basic experiments are conducted on two image classification datasets: CIFAR-100 (Krizhevsky & Hinton, 2009) and ImageNet (Russakovsky et al., 2015). This section provides the experimental details.

### A.2.1   IMAGE CLASSIFICATION ON CIFAR-100

CIFAR-100, which consists of 50,000 training images and 10,000 test images with 100 classes, is a popular classification dataset for KD research. Following the settings of CRD (Tian et al., 2020), we use 13 teacher-student pairs having either the same type or different type network architectures (see Table 1,2 in the main paper) for experiments. Each experiment is conducted for 5 separate runs, and we report top-1 mean recognition rate on the test set. For fair comparisons, we use the same training settings to conduct the experiments with our method, following CRD. Specifically, for each teacher-student pair, the model is trained by the stochastic gradient descent (SGD) optimizer for 240 epochs, with a batch size of 64, a weight decay of 0.0005 and a momentum of 0.9. The initial learning rate is set to 0.1 and decreased by a factor of 10 at epoch 150, 180 and 210.

All models are trained on an Intel Xeon Silver 4214R CPU server using one NVIDIA GeForce RTX 3090 GPU.

### A.2.2   IMAGE CLASSIFICATION ON IMAGENET

ImageNet is much more challenging than CIFAR-100, which contains over 1.2 million images for training and 50,000 images for validation, including 1,000 image classes. Following the settings of Tian et al. (2020); Yang et al. (2021); Chen et al. (2021b), we use 2 popular teacher-student pairs (see Table 3 in the main paper), namely ResNet34→ResNet18 and ResNet50→MobileNet (Howard et al., 2017), for experiments. For fair comparisons, we adopt the standard data augmentation to train and evaluate each network. For training, we first resize the input images to $256 \times 256$, then randomly sample $224 \times 224$ image crops or their horizontal flips. We standardize the cropped images with mean and variance per channel. For evaluation, we use the center crops of the resized images, and report top-1 recognition rate on the ImageNet validation set.

**Training setup for ResNet34→ResNet18.** The model is trained by SGD optimizer for 100 epochs, with a batch size of 256, a weight decay of 0.0001 and a momentum of 0.9. The initial learning rate is set to 0.1 and decreased by a factor of 10 every 30 epochs.

**Training setup for ResNet50→MobileNet.** The model is trained by SGD optimizer for 100 epochs, with a batch size of 256, a weight decay of 0.0001 and a momentum of 0.9. The initial learning rate is set to 0.1 and and scheduled to arrive at zero with a cosine decaying strategy.

All models are trained on an Intel Xeon Silver 4214R CPU server with 2 NVIDIA GeForce RTX 3090 GPUs.

### A.3 Applying NORM to more teacher-student layer pairs

Since our FT module is merely inserted after the last convolutional layer of a student network, NORM performs the proposed many-to-one representation matching between a single teacher-student layer pair. A natural question is how about the performance when applying NORM to multiple teacher-student layer pairs. In the main paper, we provide a set of ablative experiments to explore this question on ImageNet with ResNet34 as teacher and ResNet18 as student (see Table 4 in the main paper). In the experiments, we add two linear student FT modules ($N = 8$) after conv5_x (our default design) and conv4_x of ResNet18, and consider the following two settings: (1) using the same staged teacher-student representations for many-to-one matching simultaneously; (2) forcing two different staged student representations to match the teacher representation from conv5_x. For the teacher-student layer pair of conv5_x→conv4_x, we apply a $2 \times 2$ average pooling with a stride of 2 to the expanded student representation in order to match the spatial dimension of the teacher representation. Figure 5 shows an architectural overview of these two NORM variants.

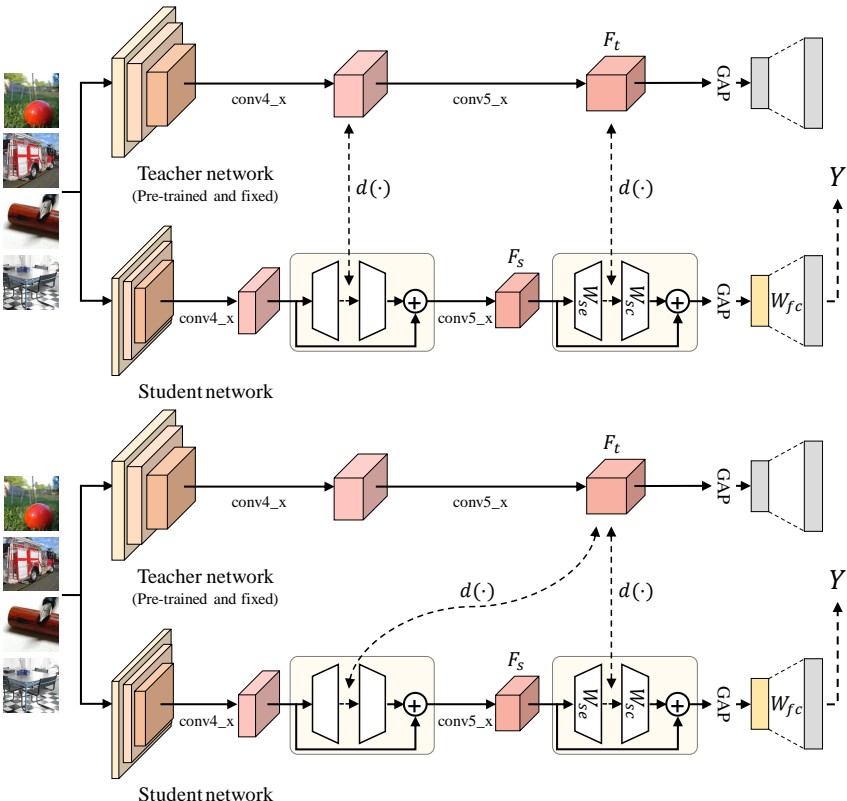

Figure 5: Architectural overview of applying NORM to two teacher-student layer pairs. For the ablative experiments on ImageNet with ResNet34 as teacher and ResNet18 as student (Table 4 in the main paper), we add two linear FT modules ($N = 8$) after conv5_x (our default design) and conv4_x of ResNet18, and consider the following two settings: (1) applying NORM to two same staged teacher-student layer pairs (after conv4_x and conv5_x) simultaneously (the figure in the first row); (2) forcing the student representations expanded from conv4_x and conv5_x to match the teacher representation from conv5_x (the figure in the second row). For the teacher-student layer pair of conv5_x→conv4_x, we apply a $2 \times 2$ average pooling with a stride of 2 to the expanded student representation in order to match the spatial dimension of the teacher representation.

### A.4 Contracting teacher representation

In our formulation, NORM enables the many-to-one representation matching via expanding the student representation to have $N$ times feature channels than the teacher's. Naturally, a reversed way for enabling the many-to-one representation matching is to contract the teacher representation to have $1/N$ times feature channels than the student's. In principle, such a reversed NORM variant is

Table 7: Comparison of expanding student representation vs. contracting teacher representation. On ImageNet with a teacher-student pair ResNet34→ResNet18, we compare NORM ($N = 8$ and $\alpha = 8$) with a reversed NORM variant which enables the many-to-one representation matching via contracting the teacher representation to have $1/N$ feature channels than the student's.

| Student | Baseline | NORM (default) | Reversed NORM |
|---|---|---|---|
| ResNet18 (w/ plain FT) | 70.13 | 71.42 | 70.82 |
| ResNet18 (w/ default FT) | 70.13 | 72.14 | 71.20 |

Table 8: Effect of different distance metrics. On CIFAR-100, we compare the performance of using 3 different distance metrics to compute the many-to-one representation matching loss of NORM ($N = 8$ and $\alpha = 10$).

| Distance metric | $l_1$-norm | $l_2$-norm (default) | MMD |
|---|---|---|---|
| ResNet32x4→ResNet8x4 | 75.10 | 76.49 | 76.62 |

more efficient than NORM. However its performance is limited by the information missing caused by dimension reduction to the teacher representation. To validate this problem, we perform a set of ablative experiments on ImageNet with ResNet34 as teacher and ResNet18 as student. We compare NORM (using a linear FT module either with or without a linear residual connection) and the reversed NORM (using a popular FT consisting of one $1 \times 1$ convolutional layer). Table 7 shows detailed results, from which we find the reversed NORM can also improve the baseline student but performs obviously worse than NORM. Specifically, NORM outperforms the reversed NORM by a top-1 margin of $0.60\%$ and $0.94\%$ for the FT module without and with a linear residual connection, respectively.

### A.5 PERFORMANCE COMPARISON WITH NETWORK RE-PARAMETERIZATION METHODS

In Table 6 of the main paper, we compare the performance of NORM with two network re-parameterization methods on the ImageNet dataset, which also apply the vanilla logits based KD during model training. For the student MobileNet, a pre-trained ResNet50 is used as the teacher in NORM, while a more powerful ResNet152 is used as the teacher in the paper of ExpandNets (Guo et al., 2020b) when performing the logits based KD. Even with a less powerful teacher model, the MobileNet model trained by our NORM without the logits based KD is obviously more accurate than that trained by ExpandNets+KD, showing $3.79\%$ top-1 gain. For the student ResNet50-1/4, we follow basic training settings of WIN (Zhou et al., 2020). Specifically, a pre-trained ResNet50 is used as our teacher. The student model is trained by SGD optimizer for 100 epochs, with a batch size of 256, a weight decay of 0.0001 and a momentum of 0.9. The initial learning rate is set to 0.1 and scheduled to arrive at zero with a cosine decaying strategy. Strong augmentations like label smoothing (Szegedy et al., 2016) and mixup (Zhang et al., 2018a) are not used in our method. After training, for the ResNet50-1/4 (student), NORM without KD reaches $68.03\%$ top-1 accuracy with a pre-trained ResNet50 (teacher), while WIN (Zhou et al., 2020) with KD gets $67.50\%$ top-1 accuracy using a pre-trained wide teacher which is 4 times larger than the student.

### A.6 EFFECT OF DIFFERENT DISTANCE METRICS

In NORM, we use the $l_2$-norm distance metric to compute the many-to-one representation matching loss $L_{norm}$ defined in Eq. 3 of the main paper. In Table 8, we compare the performance of our method with three different distance metrics including $l_2$-norm (our choice), $l_1$-norm and mean maximum discrepancy (MMD). In the experiments, we use the teacher-student pair of ResNet32x4→ResNet8x4 with $N = 8$ and $\alpha = 10$. Comparatively, $l_2$-norm is superior to $l_1$-norm. With MMD, NORM reaches $76.62\%$ accuracy for the student model, showing $0.13\%$ improvement to $l_2$-norm. This indicates that, with a better distance metric, our method might yield a student model with higher accuracy. We currently choose the $l_2$-norm distance metric owing to its simplicity and effectiveness.

### A.7 TRAINING COST COMPARISON

Table 9 shows a comparison of the total training cost of NORM and the individual training. We can see that the total training cost of NORM for the student ResNet18, MobileNet and ResNet50-1/4 is

Table 9: Training cost comparison. All models are trained on ImageNet with the same settings to those for Table 3 and Table 6 in the main paper, using an Intel Xeon Silver 4214R CPU server with two NVIDIA GeForce RTX 3090 GPUs.

| Teacher-student pair | Total training cost of individual training (hour) | Total training cost of NORM (hour) |
|---|---|---|
| ResNet34→ResNet18 | 24.14 | 67.11 |
| ResNet50→ResNet50-1/4 | 39.92 | 97.28 |
| ResNet50→MobileNet | 30.33 | 72.52 |

$2.78\times$, $2.48\times$, $2.39\times$ than the individual training, with the teacher network ResNet34, ResNet50 and ResNet50-1/2, respectively. These results indicate that NORM is efficient since it uses the standard teacher-student training framework in which the pre-trained teacher model runs in the forward phase of the whole training procedure. The forward time of the teacher can be saved by offline representation pre-computing. For easy implementation, we do not use it.

## A.8 VISUALIZATION RESULTS

NORM achieves the feature distillation goal with the proposed many-to-one representation matching mechanism at a single teacher-student layer pair. This means that the feature distribution of the student network would be more similar to that of the teacher network after NORM training, compared to the individual training. Under this context, it is necessary to study the learnt feature distributions with and without NORM. To this end, we use a teacher-student model pair (ResNet110→ResNet32) well trained by NORM ($N = 8$ and $\alpha = 10$) on the CIFAR-10 dataset (which is popularly used to analyze the learnt feature distributions in KD research) and all images in the validation set, and conduct a set of experiments to analyze the learnt last-layer feature distributions using t-SNE (Maaten & Hinton, 2008). Comparative visualization results are shown in Fig. 6, from which we can observe that NORM shows relatively strong feature mimicking capability. Each color denotes one image category in the learnt feature distribution.

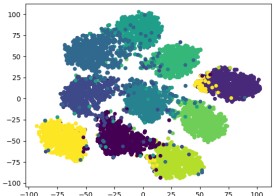 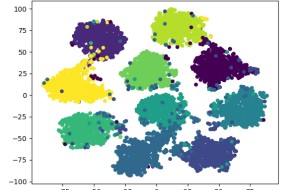 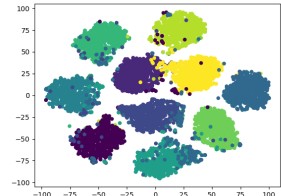

Figure 6: Comparison of learnt feature distributions. We use a well-trained ResNet110→ResNet32 model pair ($N = 8$ and $\alpha = 10$) on the CIFAR-10 dataset and all images in the validation set, and conduct a set of experiments to analyze the learnt last-layer feature distributions using t-SNE (Maaten & Hinton, 2008). The left figure shows the feature distribution from an individually trained student model; the middle figure shows the feature distribution from the student model trained by NORM; and the right figure shows the feature distribution from the pre-trained teacher model. We can observe that NORM shows relatively strong feature mimicking capability. Each color denotes one image category in the learnt feature distribution.

## A.9 MORE EXPERIMENTS AND DISCUSSIONS FOR THE REBUTTAL

In this section, we provide a lot of extra experiments and discussions provided for the rebuttal[1].

## A.9.1 A SYSTEMATIC STUDY OF NORM

So far, we have provided several sets of ablative experiments to show: (1) a large $N$ value is much better than $N = 1$, see Figure 2; (2) given the expanded student representation consisting of $N$ feature segments, the more the feature segments used to match the teacher representation, the larger the accuracy gain, see Figure 4; (3) inserting a linear FT module into different student networks usually does not bring accuracy improvement under the individual training, see Table 1, Table 2 and Table 5.

---

[1]We would like to acknowledge the contributions by intern Jiawei Fan supervised by Anbang Yao. He conducted the experiments to test the effectiveness of our method on image segmentation task, object detection task, and image classification with the masked generative learning.

Table 10: The role of the learnable ensemble layer (i.e., the second linear layer of our student FT module). On ImageNet with a teacher-student pair ResNet34→ResNet18, we compare NORM ($N = 8$ and $\alpha = 8$) with 3 different student FT designs. Best result is bolded.

| Methods | Top-1 accuracy (%) |
|---|---|
| Baseline | 70.13 |
| NORM + a default 2-layer FT module | 72.14 |
| NORM + a 2-layer FT module w/ the fixed ensemble layer | 71.91 |
| NORM + a 3-layer FT module | **72.21** |

Table 11: The role of the student FT module initialization. On ImageNet with a teacher-student pair ResNet34→ResNet18, we compare NORM ($N = 8$ and $\alpha = 8$) with 3 different weights initialization strategies. Best result is bolded.

| Methods | Top-1 accuracy (%) |
|---|---|
| Baseline | 70.13 |
| NORM (N=1) | 71.23 |
| NORM (N=8) + default weights initialization | 72.14 |
| NORM (N=8) + initialization with a higher weights variance | **72.22** |
| NORM (N=8) + initialization with the same weights to N student feature segments | 72.08 |

In order to better understand what enables the distillation effectiveness, here we systematically study the major components of NORM from the following four more aspects.

All following ablative experiments are performed on ImageNet dataset with ResNet34 as teacher and ResNet18 as student, $N = 8$ and $\alpha = 8$.

**The role of the learnable ensemble layer.** In light of the interpretation of NORM given in the Method section, the second linear layer $W_{sc}$ of our student FT module performs a learnable ensemble of $N$ distillation-augmented student feature views via fully connected channel mixing operations. Accordingly, we conduct a set of ablative experiments to explore the role of this component. In the experiments, we compare NORM with 3 different student FT designs: (a) our default two-layer student FT module, (b) a two-layer student FT module with the fixed ensemble layer (i.e., feature channels from different segments are sequentially averaged), and (c) a three-layer student FT module (i.e., having two same-size linear layers as a learnable ensemble). Detailed results are summarized in Table 10. It can be seen that all three student FT modules bring clear accuracy improvements, validating the importance of the second linear layer for ensembling $N$ augmented student feature views. Comparatively, NORM with the learnable ensemble layer is better than NORM with the fixed ensemble layer, and further improved result is attained when using a more complex learnable ensemble (two-layer).

**The role of the student FT module initialization.** Note that $N$ expanded student feature segments generated by the first linear layer of our student FT module enable the many-to-one representation matching. Next, we perform a set of ablative experiments to study the effect of different weights initialization strategies for the first linear layer. In the experiments, we compare NORM with 3 different weights initialization strategies: (a) the default weights initialization we used, (b) initialization with a higher weights variance ($8\times$ of the default), and (c) initialization with the same weights to N segments. Detailed results are summarized in Table 11. It can be seen that all three weights initialization strategies get promising performance, showing the robustness of our method to different initialization strategies. Comparatively, initialization with a higher weights variance tends to bring a bit more gain. What's more, initialization with the same weights to $N = 8$ segments is much better than $N = 1$. This benefits from the second linear layer which uses fully connected operations for dynamically mixed feature ensembling. As a result, the gradients are different to the weight groups initialized with the same values, producing different student feature segments (we also confirm this by recording and comparing them on the fly) to match the teacher representation.

**The role of the designs to construct the many-to-one representation matching.** NORM enables the many-to-one representation matching via expanding the student representation to have $N$ times feature channels than the teachers. We can also construct multiple student-to-teacher matching routes in other ways. Accordingly, we perform another set of ablative experiments to study the effect of different designs to construct the many-to-one representation matching. In the experiments, we compare 5 different designs: (a) our default design, (b) a reversed design (contracting the

Table 12: The role of the designs to construct the many-to-one representation matching. On ImageNet with a teacher-student pair ResNet34→ResNet18, we compare NORM ($N = 8$ and $\alpha = 8$) with 5 different designs to construct the many-to-one representation matching. Best result is bolded.

| Methods | Top-1 accuracy(%) |
|---|---|
| Baseline | 70.13 |
| NORM (default, student-to-teacher matching routes $N : 1$) | **72.14** |
| Reversed NORM (1 FT to contract teacher features, w/o our linear student FT module, $N : 1$) | 71.20 |
| Reversed NORM ($N$ teacher FTs, w/o our linear student FT module, $1 : N$) | 71.18 |
| Paired NORM ($N$ teacher FTs, $N$ student FTs, $N : N$) | 70.62 |
| Paired NORM ($N$ teacher FTs, w/ our linear student FT module, $N : N$) | 71.78 |

Table 13: The role of the ways to split the expanded student representation. On ImageNet with a teacher-student pair ResNet34→ResNet18, we compare NORM ($N = 8$ and $\alpha = 8$) with 3 different feature splitting strategies. Best result is bolded.

| Methods | Top-1 accuracy (%) |
|---|---|
| Baseline | 70.13 |
| NORM + sequential feature splitting (default) | 72.14 |
| NORM + random feature splitting | 72.13 |
| NORM + importance-based feature splitting | **72.15** |

teacher representation by a single FT to have $1/N$ times feature channels than the students, without using our linear student FT module), (c) a reversed design (using $N$ independent FTs to the teacher representation, without using our linear student FT module), (d) a paired design (using $N$ independent FTs to the teacher representation and another $N$ independent FTs to the student representation), and (e) a paired design (using $N$ independent FTs to the teacher representation, and using our linear student FT module). Detailed results are summarized in Table 12. It can be seen that all five designs improve the performance of the student network, validating the importance of the many-to-one representation matching concept. Comparatively, our default design achieves the best performance, and the designs that apply a single FT or multiple FTs to the teacher representation show obviously worse performance due to the information loss (that is, it is important to preserve the intact pre-trained teacher representation).

**The role of the ways to split the expanded student representation.** In NORM, for simplicity, we sequentially split the expanded student representation into $N$ feature segments. Our last set of ablative experiments is to study the effect of different feature splitting strategies. In the experiments, we test NORM with 3 different feature splitting strategies: (a) sequential feature splitting, (b) random feature splitting, and (c) importance-based feature splitting in which we first sort feature channels in descending order based on the learnt mean values of channel-wise batch normalization parameters at the fifth epoch, and then use sequential feature splitting to enable our many-to-one representation matching. Detailed results are summarized in Table 13. It can be seen that all three feature splitting strategies show almost the same performance. This is because that there is no semantic channel-wise alignment between the teacher and the expanded student representations before the many-to-one representation matching.

Based on all ablations described above, we validate the roles of the major components of our method, and provide a deep understanding of what enables the distillation effectiveness of NORM.

### A.9.2 SEMANTIC SEGMENTATION ON CITYSCAPES DATASET

In order to test the effectiveness of our method on image segmentation task, we perform two sets of experiments with Cityscapes dataset (Cordts et al., 2016). Cityscapes dataset contains 5000 images with a split of 2975, 500 and 1525 images are for training, validation and test, respectively. In the experiments, we test our method with both same type and different type teacher-student network pairs for semantic segmentation, and compare it with SKD (Liu et al., 2019a), IFVD (Wang et al., 2020), CWD (Shu et al., 2021), CIRKD (Yang et al., 2022a) and MGD (Yang et al., 2022c). Specifically, we test our method using DeepLabV3-ResNet101 (Chen et al., 2017) as teacher network and DeepLabV3-ResNet18 as student network first, and then using PSPNet-ResNet101 (Zhao et al., 2017) as teacher network and DeepLabV3-ResNet18 as student network, following the training and test settings used in very recent works of CIRKD and MGD. We add NORM ($N = 2$ for training efficiency) after the last feature layer of DeepLabV3-ResNet18. All models are trained with 8 NVIDIA Tesla

Table 14: Results comparison on Cityscapes dataset with the same type teacher-student network pair. Best result is bolded.

| Teacher | Student | Methods | Performance (mIOU, %) |
|---|---|---|---|
| DeepLabV3-ResNet101 (78.07) | DeepLabV3-ResNet18 (74.21) | SKD | 75.42 |
| DeepLabV3-ResNet101 (78.07) | DeepLabV3-ResNet18 (74.21) | IFVD | 75.59 |
| DeepLabV3-ResNet101 (78.07) | DeepLabV3-ResNet18 (74.21) | CWD | 75.55 |
| DeepLabV3-ResNet101 (78.07) | DeepLabV3-ResNet18 (74.21) | CIRKD | 76.38 |
| DeepLabV3-ResNet101 (78.07) | DeepLabV3-ResNet18 (74.21) | MGD | n/a |
| DeepLabV3-ResNet101 (78.07) | DeepLabV3-ResNet18 (74.21) | NORM (ours) | **77.03** |

Table 15: Results comparison on Cityscapes dataset with the different type teacher-student network pair. Best result is bolded.

| Teacher | Student | Methods | Performance (mIOU, %) |
|---|---|---|---|
| PSPNet-ResNet101 (78.34) | DeepLabV3-ResNet18 (73.20) | SKD | 73.87 |
| PSPNet-ResNet101 (78.34) | DeepLabV3-ResNet18 (73.20) | IFVD | n/a |
| PSPNet-ResNet101 (78.34) | DeepLabV3-ResNet18 (73.20) | CWD | 75.93 |
| PSPNet-ResNet101 (78.34) | DeepLabV3-ResNet18 (73.20) | CIRKD | n/a |
| PSPNet-ResNet101 (78.34) | DeepLabV3-ResNet18 (73.20) | MGD | 76.02 |
| PSPNet-ResNet101 (78.34) | DeepLabV3-ResNet18 (73.20) | NORM (ours) | **76.51** |

V100-SXM3 GPUs. Detailed results are summarized in Table 14 and Table 15. It can be seen that our method achieves new state-of-the-art results on these two teacher-student network pairs, validating its effectiveness in handling semantic segmentation task.

### A.9.3 OBJECT DETECTION ON MS COCO DATASET

In order to test the effectiveness of our method on object detection task, we also perform two sets of experiments with MS COCO dataset (Lin et al., 2014). MS COCO dataset (2017 version) contains 118,000 training images and 5,000 validation images with 80 object classes. In the experiments, we test our method with both same type and different type teacher-student network pairs, and compare it with FKD (Zhang & Ma, 2021), CWD (Shu et al., 2021), FGD (Yang et al., 2022b) and MGD (Yang et al., 2022c). Specifically, we test our method using RetinaNet-ResNeXt101 (Lin et al., 2017b) as teacher detector and RetinaNet-ResNet50 as student detector first, and then using Cascade Mask RCNN-ResNeXt101 (He et al., 2017) as teacher detector and Faster RCNN-ResNet50 (Ren et al., 2015) as student detector, following the training and test settings used in very recent papers of FGD and MGD. We add NORM ($N = 4$ for training efficiency) after each output of the FPN neck (Lin et al., 2017a) of RetinaNet-ResNet50/Faster RCNN-ResNet50. All models are trained with 8 NVIDIA Tesla V100-SXM3 GPUs. Detailed results are summarized in Table 16 and Table 17. It can be seen that our method achieves new state-of-the-art mAP results on these two teacher-student detector pairs, validating its effectiveness in handling object detection task.

### A.9.4 MORE DISCUSSIONS AND EXPERIMENTS

Here, we additionally discuss some potential extensions of NORM.

**NORM vs. self-supervised learning methods.** Recently, self-supervised learning research with self-distillation settings (a particular type of unsupervised learning) has attracted increasing attention. In this line of research (e.g., SimCL (Chen et al., 2020b), MoCo (He et al., 2020), SwAV (Caron et al., 2020), BYOL (Grill et al., 2020), DenseCL (Wang et al., 2021), DINO (Caron et al., 2021), iGPT (Chen et al., 2020a), iBOT (Zhou et al., 2022), BeiT (Bao et al., 2022) and MAE (He et al., 2022)), matching $N$ features of the student to $N$ features of the teacher by referring to dense features of different image patches has been explored. Indeed, they also have the concept of many parallel

Table 16: Results comparison on MS COCO dataset with the same type teacher-student network pair. Best result is bolded.

| Teacher | Student | Methods | mAP (%) | $AP_S$ (%) | $AP_M$ (%) | $AP_L$ (%) |
|---|---|---|---|---|---|---|
| RetinaNet-ResNeXt101 (41.0) | RetinaNet-ResNet50 (37.4) | FKD | 39.6 | 22.7 | 43.3 | 52.5 |
| RetinaNet-ResNeXt101 (41.0) | RetinaNet-ResNet50 (37.4) | CWD | 40.8 | 22.7 | 44.5 | 55.3 |
| RetinaNet-ResNeXt101 (41.0) | RetinaNet-ResNet50 (37.4) | FGD | 40.7 | 22.9 | 45.0 | 54.7 |
| RetinaNet-ResNeXt101 (41.0) | RetinaNet-ResNet50 (37.4) | MGD | 41.0 | **23.4** | 45.3 | 55.7 |
| RetinaNet-ResNeXt101 (41.0) | RetinaNet-ResNet50 (37.4) | NORM (ours) | **41.1** | 23.3 | **45.3** | **55.7** |

Table 17: Results comparison on MS COCO dataset with the different type teacher-student network pair. Best result is bolded.

| Teacher | Student | Methods | mAP (%) | $AP_S$ (%) | $AP_M$ (%) | $AP_L$ (%) |
|---|---|---|---|---|---|---|
| Cascade Mask RCNN-ResNeXt101 (47.3) | Faster RCNN-ResNet50 (38.4) | FKD | 41.5 | 23.5 | 45.0 | 55.3 |
| Cascade Mask RCNN-ResNeXt101 (47.3) | Faster RCNN-ResNet50 (38.4) | CWD | 41.7 | 23.3 | 45.5 | 55.5 |
| Cascade Mask RCNN-ResNeXt101 (47.3) | Faster RCNN-ResNet50 (38.4) | FGD | 42.0 | 23.8 | 46.4 | 55.5 |
| Cascade Mask RCNN-ResNeXt101 (47.3) | Faster RCNN-ResNet50 (38.4) | MGD | 42.1 | 23.7 | 46.4 | 56.1 |
| Cascade Mask RCNN-ResNeXt101 (47.3) | Faster RCNN-ResNet50 (38.4) | NORM (ours) | **42.4** | **24.0** | **46.6** | **56.3** |

Table 18: Results comparison on ImageNet dataset with a teacher-student pair of vision transformer architectures. We combine our NORM with the masked generative learning proposed in VITKD. Best result is bolded.

| Teacher | Student | Methods | Top-1 accuracy (%) |
|---|---|---|---|
| DeiT-Small(80.69) | DeiT-Tiny (74.42) | KD | 75.01 |
| DeiT-Small(80.69) | DeiT-Tiny (74.42) | NKD | 75.48 |
| DeiT-Small(80.69) | DeiT-Tiny (74.42) | VITKD | 75.40 |
| DeiT-Small(80.69) | DeiT-Tiny (74.42) | VITKD+NKD | 76.18 |
| DeiT-Small(80.69) | DeiT-Tiny (74.42) | VITKD+NORM (ours) | **76.55** |

matching routes with their corresponding components, but our method differs with them in focus, formulation and application. Firstly, existing methods of self-supervised learning with self-distillation settings aim to learn a proper visual representation of a single backbone from a large set of unlabeled images, while our work aims to improve the visual representation of a small student network by a pre-trained larger teacher network under condition that both of them are trained on a set of labelled images. Secondly, in formulation, many existing self-supervised learning methods (e.g., SimCL, MoCo, SwAV, BYOL, DenseCL and DINO) define a contrastive predication task that encourages the encoder (backbone) to attract similar (positive) sample views and dispel different (negative) sample views with their corresponding contrastive losses leveraging data augmentation, and some others (e.g., iGPT, BeiT and MAE) define a masked reconstruction task that uses a decoder to predict the original image (pixel-wise/patch-wise) given the representation learnt by the encoder (backbone) with their corresponding reconstruction losses leveraging masked image modeling. iBOT further combines masked image modeling into self-supervised contrastive learning for improved performance. To these methods such as DenseCL and iBOT, self-distillation is performed between the target model (student) and the online model (teacher, an exponential moving average of student parameters) via contrasting dense sample view pairs of the whole image or the patch (some patches may be masked), and both models share the same encoder (backbone). In contrast, our method leverages a linear FT module added after the last convolutional layer of a small student network to formulate many-to-one representation matching scheme between a single teacher-student layer via student feature expansion and splitting, and dense mimicking. All segments of our expanded student features are simultaneously forced to be similar to the teacher features, and there is no contrastive matching and no paired data augmentation to the input image. Thirdly, in application, the trained encoder of self-supervised methods is typically used to downstream tasks, while the efficient student network trained by our method is for direct deployment. Besides, iBOT, iGPT, BeiT and MAE with masked image modeling are particularly used to train vision transformers in unsupervised learning regime, while our method is mainly for training efficient convolutional neural networks in supervised learning regime.

Note that recent work CRD extends contrastive learning into KD research by presenting a novel contrastive distillation loss, and in the main paper we already validated the effectiveness of combining our method with CRD (see Table 1 and Table 2).

**Combining NORM with masked generative learning.** In the main paper, we evaluate our method with 16 teacher-student network pairs (13/3 pairs on CIFAR-100/ImageNet dataset). The main reason for this is that in the community, mainstream KD methods for image classification, e.g., 19 recent KD methods compared in our work, typically use convolutional architectures for performance evaluation. We follow them for fair and easy comparisons. Benefited from the simplicity of our method, we can easily use it to other network architectures. We notice that a very recent work VITKD (Yang et al., 2022d) explores the use of the aforementioned masked generative learning for vision transformer based KD research. Benefited from the simplicity of our method, we can easily combine NORM with VITKD to test our methods effectiveness in training a vision transformer under masked generative learning. Accordingly, we perform a set of new experiments on ImageNet dataset, using DeiT-Small (Touvron et al., 2021) as teacher network and DeiT-Tiny as student network, following the settings of VITKD. For the student network, we add NORM ($N = 4$ for training

efficiency) to the same layers as VITKD. Detailed results are summarized in Table 18. It can be seen that our method also works well on transformer architectures, achieving promising results. Here, it should be noted that the standard training of a popular transformer model is very time consuming due to significantly larger model size and longer training schedule, compared to that of mainstream convolutional networks (e.g., 300 epochs for DeiTs vs. 100 epochs for ResNets). One run of training the above teacher-student transformer pair by NORM needs about 2.5 days with 8 NVIDIA Tesla V100-SXM3 GPUs. We will continue to explore the potential of our method to more teacher-student transformer pairs in the future.

## A.10 LIMITATIONS OF NORM

Despite of its simple formulation and effectiveness, NORM has two limitations. The major limitation is applying the many-to-one representation matching to multiple teacher-student layer pairs cannot achieve further improved distillation performance, unlike many existing KD methods. This is due to the linear property of our student FT design, as we explored in the ablative experiments (see Table 4 and Table 5 in the main paper). Besides, applying the many-to-one representation matching to multiple teacher-student layer pairs will lead to increased training cost because the many-to-one representation matching also needs to be computed multiple times.

