# OpenReview forum: "NORM: Knowledge Distillation via N-to-One Representation Matching"
_ICLR.cc/2023/Conference — ICLR 2023 poster_

### Official Review · Reviewer_iHQs · 2022-10-18

**Confidence:** 4
**Correctness:** 3
**Technical Novelty And Significance:** 2
**Empirical Novelty And Significance:** 3
**Recommendation:** 6

**Clarity, Quality, Novelty And Reproducibility:**

Clarity: the paper is well written and very easy to understand. Everything is very clear.

Reproducibility: the authors promise to release code for this work. Additionally, given that the method is very simple, I don’t see any potential problems with reproducibility.

Novelty: the method builds on top of standard techniques in the area of feature distillation, and the main contribution is a simple modification to the traditional two-stage distillation framework. There is some novelty, although it is not so strong.

Quality: even though the method is simple and not so novel, the paper is very well written and the experiments quite comprehensive. The execution quality is generally high, but one aspect that is lacking is the justification of the method, as I discuss in detail in the “Weaknesses” section above.

**Strength And Weaknesses:**

Strengths:

S1) Knowledge distillation is an important and active topic in the community, which is addressed in this work.

S2) The paper is generally well written and easy to follow.

S3) The method is very simple and easy to understand.

S4) The experimental evaluation is significant, considering many different network architectures and several baselines. Ablations are performed to understand which parts of the system matter more.

Weaknesses:

W1) The main concern I have with this paper is that we don’t really understand why the proposed method works. Simply create multiple heads with the exact same shape and distill the teacher into each of them simultaneously. Why is this supposed to be better than simply having one head here? I do recognize that the authors provide many experiments indicating that this replication of heads apparently really helps, which to me feels surprising.

In general, I think there should be more discussion on this. For example, what if all N heads are initialized with the exact same random weights? If this is the case, I believe that would degenerate into the single-head (N=1) case? What types of initialization are needed here? Maybe we need higher initialization variance in order to capture different optimization routes?

W2) A question: can you please clarify if the networks are using global average pooling? Or if the fully connected layer from the classifier is applied on top of the final feature map directly? Can you please clarify additionally if the baseline methods you are comparing to also use global average pooling, or if in their cases the FC layer is applied on top of the final feature map directly? I wanted to make sure everything is comparable in this respect, and the gains we see are not coming from the removal of the global pooling, for example.

Minor details:

M1) Section 3.1 does not concern new contributions from this paper. I suggest the authors revise its title to “Background: One-to-one representation matching”, to emphasize the fact that background is presented.

M2) In several parts of the manuscript the word “typed” is used incorrectly, eg “same typed”, “different typed”. In most cases it should be “type”.


**Summary Of The Paper:**

The paper proposes a new knowledge distillation method called NORM, based on the idea of many-to-one representation matching. The method follows a two-stage distillation process using a linear Feature Transform (FT) module. The FT module has two layers: the first expands the number of channels to N times those from the teacher, and the second shrinks the number of channels back to the original student feature space. The expanded representation is forced to approximate the teacher representation N times. Experiments are conducted on the CIFAR100 and ImageNet datasets with various teacher-student pairs.

**Summary Of The Review:**

The paper tackles an important problem, proposing a modification to the standard two-stage distillation paradigm. Experiments are comprehensive and seem to show improved performance in common benchmarks, but the method lacks justification.

---

> ### Author Response · Authors · 2022-11-19
> **Responses to Official Review by Reviewer iHQs: Part 4**
>
> 3.**To your minor comment M1** “Section 3.1 does not concern new contributions from this paper. I suggest the authors revise its title to “Background: One-to-one representation matching”, to emphasize the fact that background is presented.”
>
> **Our responses** are: **(1)** Yes, Section 3.1 does not concern new contributions. It is mainly for reviewing the representation matching of existing two-stage feature distillation (FD) methods in a general formulation. By this, we reveal that existing FD methods adopt the one-to-one representation matching at any pre-selected teacher-student layer pair, and then present the basic motivation of our work. More specifically, existing FD methods perform global single-shot feature mimicking process, meaning that there exists only one knowledge transfer route between any teacher-student layer pair. We conjecture that if we can formulate an effective mechanism to introduce multiple global knowledge transfer routes between any teacher-student layer pair, it will offer more chances to inject the intact teacher’s knowledge into the student network, and improved FD performance could be attained; **(2)** Following your kind advice, in the revised manuscript, the title of Section 3.1 is revised to “Background: One-to-one representation matching”.
>
> 4.**To your minor comment M2** “In several parts of the manuscript the word “typed” is used incorrectly, eg “same typed”, “different typed”. In most cases it should be “type””.
>
> **Our response** is: following your careful advice, all related places of the word “typed” are corrected in the revised manuscript.
>
> 5.**To your comment** “Reproducibility: the authors promise to release code for this work. Additionally, given that the method is very simple, I don’t see any potential problems with reproducibility.”
>
> **Our responses** are: **(1)** Thanks for recognizing the simplicity of our method. **Now, in the Supplementary Material**, we provide a clean version of our source code for main image classification experiments on both CIFAR-100 and ImageNet datasets; **(2)** Next, we will further improve and clean our newly added code for semantic segmentation and objection detection tasks, which will be also released later. We hope it could help the community to advance KD research.
>
> **Finally**, regarding more experiments and discussions that we have made during the rebuttal phase, you are referred to our top-level comments titled **“The Summary of Our Responses to All Official Reviews”**, our responses to the other reviewers, and the revised manuscript.

---

> > ### Comment · Reviewer_iHQs · 2022-12-05
> > **Thank you for the comprehensive rebuttal**
> >
> > I'd like to thank the authors for the comprehensive rebuttal. The authors added lots of experiments and discussions, and improved the manuscript based on my comments.
> >
> > The contribution seems moderately significant overall. In a way, it still feels more like a "trick" to make these distillation methods work better, but it is comprehensively evaluated and discussed. We still don't know exactly why it works, but it seems clear that it works.
> >
> > Based on this, I will upgrade my score to "marginally above the acceptance threshold".

---

> > > ### Author Response · Authors · 2022-12-05
> > > **Thanks for the Recognition of Our Rebuttal**
> > >
> > > Thank you so much for the recognition of our responses. We are glad to see that you have raised your score.
> > >
> > > We will make more analysis to improve our paper further.
> > >
> > > Many thanks for your constructive comments, time and patience.

---

> ### Author Response · Authors · 2022-11-19
> **Responses to Official Review by Reviewer iHQs: Part 3**
>
> **(iv) The role of the ways to split the expanded student representation.** In NORM, for simplicity, we sequentially split the expanded student representation into $N$ feature segments. Our last set of ablative experiments is to study the effect of different feature splitting strategies. In the experiments, we test NORM with 3 different feature splitting strategies: (a) sequential feature splitting, (b) random feature splitting, and (c) importance-based feature splitting is which we first sort feature channels in descending order based on the learnt mean values of channel-wise batch normalization parameters at the fifth epoch, and then use sequential feature splitting to enable our many-to-one representation matching. Detailed results are summarized in the below Table. It can be seen that all three feature splitting strategies show almost the same performance. This is because that there is no semantic channel-wise alignment between the teacher and the expanded student representations before the many-to-one representation matching.
>
> |Methods| Top-1 accuracy (%)|
> |--|:--:|
> |Baseline|70.13|
> |NORM + sequential feature splitting (default)|72.14|
> |NORM + random feature splitting|72.13|
> |NORM + importance-based feature splitting|**72.15**|
>
> **Based on all ablative experiments described above, we validate the roles of the major components of our method, and provide a deep understanding of what enables the distillation effectiveness of NORM.**
>
> **More Experiments:** (1) We also provide multiple sets of experiments on Cityscapes and MS COCO datasets to show the effectiveness of our method in handling semantic segmentation and object detection tasks. Again, our method attains leading performance. **You are referred to our responses to reviewer Xp75 for detailed comparisons**; (2) We also provide a set of experiments on ImageNet dataset to show the effectiveness of our method in training transformer architectures, achieving promising results. **You are referred to our responses to reviewer d7Sn for detailed comparisons.**
>
> 2.**To W2, your question regarding the use of the global average pooling or the fully connected layer** “can you please clarify if the networks are using global average pooling? Or if the fully connected layer from the classifier is applied on top of the final feature map directly? Can you please clarify additionally if the baseline methods you are comparing to also use global average pooling, or if in their cases the FC layer is applied on top of the final feature map directly? I wanted to make sure everything is comparable in this respect, and the gains we see are not coming from the removal of the global pooling, for example.”
>
> **Our responses** are: **(1)** For all 13 teacher-student network pairs tested on CIFAR-100 dataset and 3 teacher-student network pairs tested on ImageNet dataset, there is a global average pooling (GAP) after the last convolutional layer of each student (/teacher) network, followed with a fully connected (FC) layer. This is the common design to modern convolutional neural networks (CNNs). That is, we always use public CNN architectures in Pytorch for experiments, following the settings of the seminal work CRD (Tian et al., ICLR 2020) as stated in the original manuscript; **(2)** In our method, the linear student feature transform (FT) module is added after the last convolutional layer, and the original GAP is remained before the FC layer. Owing to the linear property of our student FT module, the GAP and the FC layer, after training, our student FT module will be directly merged into its subsequent FC layer, and the position of the GAP is not changed, making no architectural modifications to the original student network at inference; **(3)** Actually, in the original manuscript, we already described these points **in the paragraph “Implementation and Inference” of Section 3.2** and provided the proofs of them in **the Appendix (Section A.1)**; **(4)** To further improve the clarity, in the revised manuscript, we add the GAP into Figure 1 and Figure 5; **(5)** Existing feature distillation methods typically use their FT designs as auxiliary branches which will be discarded after training, so these baselines also do not modify original student network architectures at inference; **(6)** In this respect and given the same benchmark settings, all experimental comparisons are fair, and it is clear that the gains of our method are from the proposed many-to-one representation matching.

---

> ### Author Response · Authors · 2022-11-19
> **Responses to Official Review by Reviewer iHQs: Part 2**
>
> **(ii) The role of the student FT module initialization.** Note that $N$ expanded student feature segments generated by the first linear layer of our student FT module enable the many-to-one representation matching. Next, we perform a set of ablative experiments to study the effect of different weights initialization strategies for the first linear layer, following your truly insightful suggestions. In the experiments, we compare NORM with 3 different weights initialization strategies: (a) the default weights initialization we used in Pytorch, (b) initialization with a higher weights variance (8$\times$ of the default), and (c) initialization with the same weights to N segments. Detailed results are summarized in the below Table. It can be seen that all three weights initialization strategies get promising performance, showing the robustness of our method to different initialization strategies. Comparatively, initialization with a higher weights variance tends to bring a bit more gain. What’s more, initialization with the same weights to $N=8$ segments is much better than $N=1$. This benefits from the second linear layer which uses fully connected operations for dynamically mixed feature ensembling. As a result, the gradients are different to the weight groups initialized with the same values, producing different student feature segments (we also confirm this by recording and comparing them on the fly) to match the teacher representation.
>
> |Methods|Top-1 accuracy (%)|
> |--|:--:|
> |Baseline|70.13|
> |NORM (N=1)|71.23|
> |NORM (N=8) + default weights initialization |72.14|
> |NORM (N=8) + initialization with a higher weights variance|**72.22**|
> |NORM (N=8) + initialization with the same weights to N student feature segments|72.08|
>
> **(iii) The role of the designs to construct the many-to-one representation matching.** NORM enables the many-to-one representation matching via expanding the student representation to have $N$ times feature channels than the teacher’s. We can also construct multiple student-to-teacher matching routes in other ways. Accordingly, we perform another set of ablative experiments to study the effect of different designs to construct the many-to-one representation matching, following the insightful suggestions by Reviewer xShr. In the experiments, we compare 5 different designs: (a) our default design, (b) a reversed design (contracting the teacher representation by a single FT to have $1/N$ times feature channels than the student’s, without using our linear student FT module), (c) a reversed design (using $N$ independent FTs to the teacher representation, without using our linear student FT module), (d) a paired design (using $N$ independent FTs to the teacher representation and another $N$ independent FTs to the student representation), and (e) a paired design (using $N$ independent FTs to the teacher representation, and using our linear student FT module). Detailed results are summarized in the below Table. It can be seen that all five designs improve the performance of the student network, validating the importance of the many-to-one representation matching concept. Comparatively, our default design achieves the best performance, and the designs that apply a single FT or multiple FTs to the teacher representation show obviously worse performance due to the information loss (that is, it is important to preserve the intact pre-trained teacher representation).
>
> |Methods|Top-1 accuracy(%)|
> |--|:--:|
> |Baseline|70.13|
> |NORM (default, student-to-teacher matching routes $N:1$) |**72.14**|
> |Reversed NORM (1 FT to contract teacher features, w/o our linear student FT module, $N:1$) |70.82|
> |Reversed NORM ($N$ teacher FTs, w/o our linear student FT module, $1:N$)|71.18|
> |Paired NORM ($N$ teacher FTs, $N$ student FTs, $N:N$)|70.62|
> |Paired NORM ($N$ teacher FTs, w/ our linear student FT module, $N:N$)|71.78|

---

> ### Author Response · Authors · 2022-11-19
> **Responses to Official Review by Reviewer iHQs: Part 1**
>
> Thank you so much for the detailed and constructive comments, and the recognition of the problem we addressed, the novelty of our proposed method, the writing, the experimental evaluation and ablations. Please see our below responses to your concerns and questions one by one.
>
> 1.**To W1, your main concern regarding a deep discussion on where the effectiveness of our many-to-one representation matching method comes from** “The main concern I have with this paper is that we don’t really understand why the proposed method works. Simply create multiple heads with the exact same shape and distill the teacher into each of them simultaneously. Why is this supposed to be better than simply having one head here? I do recognize that the authors provide many experiments indicating that this replication of heads apparently really helps, which to me feels surprising. In general, I think there should be more discussion on this. For example, what if all N heads are initialized with the exact same random weights? If this is the case, I believe that would degenerate into the single-head (N=1) case? What types of initialization are needed here? Maybe we need higher initialization variance in order to capture different optimization routes?”
>
> **Our responses consist of two Parts**:
>
> **Part 1: Interpretation of NORM**, the formulation of NORM can be interpreted as a way of learning a dynamically mixed feature ensemble over multiple augmented views of the same student representation by forcing them to mimic the intact teacher representation simultaneously. More precisely, in NORM, the first linear layer $W_{se}$ of our student feature transform (FT) module acts as a set of independently initialized feature transforms to generate $N$ channel-expanded views of the student feature $F_s$ whose representation abilities are then parallelly augmented by the distillation supervision from the intact teacher feature $F_t$, and the second linear layer $W_{sc}$ performs a learnable ensemble of $N$ distillation-augmented student feature views via fully connected channel mixing operations (which make the feature ensemble $F_{sc}$ has the same size to $F_s$, guaranteeing the absorbable property of NORM at inference). Our mixed student feature ensemble learning further benefits from the standard cross-entropy loss supervised by the ground truth labels. Accordingly, in Section 3.2 of the revised manuscript we add **a paragraph titled “Interpretation of NORM”**.
>
> **Part 2: A systematic study of NORM**, recall that in the original manuscript, we provide several sets of ablative experiments to show: (1) a large $N$ value is much better than $N=1$, see Figure 2; (2) Given the expanded student representation consisting of $N$ feature segments, the more the feature segments used to match the teacher representation, the larger the accuracy gain, see Figure 4; (3) Inserting a linear FT module into different student networks usually does not bring accuracy improvement under individual training, see Table 1, Table 2 and Table 5. **In order to better understand what enables the distillation effectiveness, here we systematically study the major components of NORM from the following four more aspects:**
>
> **Basic experimental settings:** All following ablative experiments are performed on ImageNet dataset with ResNet34 as teacher and ResNet18 as student, $N=8$ and $\alpha=8$ (i.e., the default settings in the original manuscript).
>
> **(i) The role of the learnable ensemble layer.** In light of the above interpretation of NORM, the second linear layer $W_{sc}$ of our student FT module performs a learnable ensemble of $N$ distillation-augmented student feature views via fully connected channel mixing operations. Accordingly, we conduct a set of ablative experiments to explore the role of this component. In the experiments, we compare NORM with 3 different student FT designs: (a) our default two-layer student FT module, (b) a two-layer student FT module with the fixed ensemble layer (i.e., feature channels from different segments are sequentially averaged), and (c) a three-layer student FT module (i.e., having two same-size linear layers as a learnable ensemble). Detailed results are summarized in the below Table. It can be seen that all three student FT modules bring clear accuracy improvements, validating the importance of the second linear layer for ensembling $N$ augmented student feature views. Comparatively, NORM with the learnable ensemble layer is better than NORM with the fixed ensemble layer, and further improved result is attained when using a more complex learnable ensemble (two-layer).
>
> |Methods|Top-1 accuracy (%)|
> |--|:--:|
> |Baseline|70.13|
> |NORM + a default 2-layer student FT module|72.14|
> |NORM + a 2-layer student FT module w/ the fixed ensemble layer|71.91|
> |NORM + a 3-layer student FT module|**72.21**|

---

### Official Review · Reviewer_xShr · 2022-10-21

**Confidence:** 4
**Correctness:** 3
**Technical Novelty And Significance:** 3
**Empirical Novelty And Significance:** 2
**Recommendation:** 6

**Clarity, Quality, Novelty And Reproducibility:**

As detailed above, the paper is well written and the method pretty clear except for the distillation loss. The idea is somewhat novel.

**Strength And Weaknesses:**

Strengths:
- Well written
- Mostly clear
- Somewhat novel

Weaknesses:
- Too narrow experimental setting
- Not particularly effective

The paper is well written and the method is mostly clear except for the distillation loss.

In particular, the default is Euclidean distance between features (3), referred to as NORM. In the experiments, the "vanilla logits based distillation" is additionally used, referred to as NORM+KD, citing the original distillation paper (Hinton et al. 2015).

But the original distillation loss in (Hinton et al. 2015) is cross entropy with soft targets given by the teacher, whereas the Euclidean distance between logits is only discussed as an approximation when the temperature is high. So the "vanilla logits based distillation" is not clear. It should be given formally as a second option below (3). This is very important given that NORM+KD is the best performing option. It cannot remain undefined.

The names NORM and NORM+KD are also not appropriate, because it is implied that NORM does not include distillation; which is not true as (3) is a form of distillation. It is more appropriate to use NORM+A and NORM+B for some A and B.

The novelty appears to be ok as I am not aware of such trick for knowledge distillation. However, a similar trick exists of introducing a number of classifiers by Littwin and Wolf 2015, "The Multiverse Loss for Robust Transfer Learning." The differences are that we have N classifiers rather than N subtensors, C classes rather than C channels, cross-entropy rather than distillation loss and an additional constraint that the classifiers are orthogonal. But structurally, the ideas are very similar. Some discussion is in order.

Another very related idea is to match N features of the student to N features of the teacher by referring to dense features of different image patches. This has been explored in self-distillation settings for self-supervision, e.g.
(Dense CL) Wang et al. 2021. Dense Contrastive Learning for Self-Supervised Visual Pre-Training.
(iBOT) Zhou et al. 2022. iBOT: Image BERT Pre-Training with Online Tokenizer.
Such dense matching is an alternative way "to introduce many parallel knowledge transfer routes." A discussion is in order. A comparison would be a great addition.

The paper includes extensive experiments on several teacher-student model pairs and a lot of ablation studies. However:
- In Table 1, NORM+KD beats the competition only in two columns where many or all of the competitors are missing. In Table 2, it is best in more columns but again many of the competitors are missing. I do not find these results conclusive.
- More importantly, two-stage knowledge-distillation from a larger to a smaller network for supervised classification is too narrow. For this architectural modification to be properly validated, I would suggest extending to online self-distillation, self-supervised and semi-supervised settings, continual learning, vision transformers and other tasks like object detection. It would be particularly interesting to combine with methods like BYOL, DINO, iBOT, Dense CL etc., which use self-distillation.

An ablation of a "reverse" idea in addition to "Contracting teacher representation" would be to match 1 representation of the student to N representations of the teacher. N-to-N could also be explored.

Appendix A.1 appears to be an overkill. The composition of linear layers is obviously linear.

My final rating will depend on how many additional experiments can be done and how effective NORM will be shown to be.

**Summary Of The Paper:**

This paper proposes an architectural trick for two-stage knowledge-distillation from a large to a small network. It expands the student feature tensor of the last layer from C to NC channels, applies distillation from the teacher feature tensor to each of the N sub-tensors, then contracts back to the original feature dimensions, C. Both mappings are linear and at inference they are merged into a single CxC layer.

**Summary Of The Review:**

This is an interesting idea. The distillation loss is unclear but I am sure this can be corrected. However, the experimental setting is too narrow and the method does not appear to be very effective.

**Update**: The rebuttal addresses my concerns to an extent and although further improvement is possible, I raise my recommendation from 5 to 6.

---

> ### Author Response · Authors · 2022-11-18
> **Responses to Official Review by Reviewer xShr: Part 6**
>
> |Teacher|Student|Distillaton Methods|Performance (mIOU, %)|
> |--|:--:|:--:|:--:|
> |PSPNet-ResNet101 (78.34)|DeepLabV3-ResNet18 (73.20)|SKD [1]|73.87|
> |PSPNet-ResNet101 (78.34)|DeepLabV3-ResNet18 (73.20)|IFVD [2]|n/a|
> |PSPNet-ResNet101 (78.34)|DeepLabV3-ResNet18 (73.20)|CWD [3]|75.93|
> |PSPNet-ResNet101 (78.34)|DeepLabV3-ResNet18 (73.20)|CIRKD [4]|n/a|
> |PSPNet-ResNet101 (78.34)|DeepLabV3-ResNet18 (73.20)|MGD [5]|76.02|
> |PSPNet-ResNet101 (78.34)|DeepLabV3-ResNet18 (73.20)|NORM (ours)|**76.51**|
>
> [1] Yifan Liu, et al., “Structured knowledge distillation for semantic segmentation”, CVPR 2019
>
> [2] Yukang Wang, et al., “Intra-class feature variation distillation for semantic segmentation”, ECCV 2020
>
> [3] Changyong Shu, et al., “Channel-wise knowledge distillation for dense prediction”, ICCV 2021.
>
> [4] Chuanguang Yang, et al., "Cross-image relational knowledge distillation for semantic segmentation", CVPR 2022.
>
> [5] Zhendong Yang, et al., "Masked Generative Distillation", ECCV 2022.
>
> **Study the combination of NORM with the masked generative learning on ImageNet dataset.** We notice that a very recent work VITKD [3] (posted on Arxiv in September 2022) explores the use of recent popular masked generative learning (e.g., iGPT, BeiT, iBOT and MAE) for vision transformer based KD research. Benefited from the simplicity of our method, we can easily combine NORM with VITKD to test our method’s effectiveness in training a vision transformer under masked generative learning. Accordingly, we perform a set of new experiments on ImageNet dataset, using DeiT-Small as teacher network and DeiT-Tiny as student network, following the settings of VITKD. For the student network, we add NORM ($N=4$ for training efficiency) to the same layers as VITKD. Detailed results are summarized in the below Table. It can be seen that our method also works well on transformer architectures, achieving promising results. Here, it should be noted that the standard training of a popular transformer model is very time consuming due to significantly larger model size and longer training schedule, compared to that of mainstream convolutional networks (e.g., 300 epochs for DeiTs vs. 100 epochs for ResNets). One run of training the above teacher-student transformer pair by NORM needs about 2.5 days with 8 Nvidia Tesla V100-SXM3 GPUs. We will continue to explore the potential of our method to more teacher-student transformer pairs in the future.
>
> |Teacher|Student|Distillaton Methods|Top-1 accuracy (%)|
> |--|:--:|:--:|:--:|
> |DeiT-Small(80.69)|DeiT-Tiny (74.42)|KD [1]|75.01|
> |DeiT-Small(80.69)|DeiT-Tiny (74.42)|NKD [2]|75.48|
> |DeiT-Small(80.69)|DeiT-Tiny (74.42)|VITKD [3]|75.40|
> |DeiT-Small(80.69)|DeiT-Tiny (74.42)|VITKD+NKD|76.18|
> |DeiT-Small(80.69)|DeiT-Tiny (74.42)|VITKD+NORM (ours)|**76.55**|
>
> [1] Geoffrey Hinton, et al, “Distilling the knowledge in a neural network”, arXiv preprint arXiv:1503.02531, 2015.
>
> [2] Zhendong Yang, et al., "Rethinking knowledge distillation via cross-entropy", arXiv preprint arXiv: 2208.10139, 2022.
>
> [3] Zhendong Yang, et al., "ViTKD: Practical Guidelines for ViT feature knowledge distillation", arXiv preprint arXiv:2205.01529, 2022.
>
> 9.**To your general comments** “Too narrow experimental setting” and “Not particularly effective”.
>
> **Our responses** are: **(1)** Considering the experiments that we now have done on the tasks of image classification (including CIFAR-100 with 13 teacher-student pairs and ImageNet with 3 teacher-student pairs), semantic segmentation (Cityscapes with 2 teacher-student pairs) and object detection (MS COCO with 2 teacher-student pairs), your concern on narrow experimental setting should be well addressed; **(1)** Considering our method mostly show leading performance in all above test cases, your concern on our method’s effectiveness should also be well addressed.
>
> **Finally**, regarding more experiments and discussions that we have made during the rebuttal phase, you are referred to our top-level comments titled **“The Summary of Our Responses to All Official Reviews”**, our responses to the other reviewers, and the revised manuscript.

---

> > ### Comment · Reviewer_xShr · 2022-11-23
> > **Feedback on responses part 6**
> >
> > I acknowledge this is a successful extension in one more of the suggested directions, that of vision transformers.
> >
> > Overall, although the paper could be improved more, in particular e.g. by extending to self-distillation as discussed in part 3 and by reproducing SoTA methods to complete tables as discussed in part 4, I acknowledge that an impressive amount of work has been done during rebuttal and I raise my recommendation accordingly from 5 to 6.
> >
> > I strongly recommend further improvement according to my suggestions in the camera-ready version in case the paper is accepted.

---

> > > ### Author Response · Authors · 2022-11-23
> > > **Responses to “Feedback on responses part 6”**
> > >
> > > Thank you so much for the recognition of our responses. We are glad to see that you have raised your score.
> > >
> > > We will continue to improve experimental comparisons, discussions, and etc., so as to further improve our paper during the final paper revision.
> > >
> > > Many thanks for your constructive comments, time and patience.

---

> ### Author Response · Authors · 2022-11-18
> **Responses to Official Review by Reviewer xShr: Part 5**
>
> 8.**To your suggestions regarding the experiment improvements** “More importantly, two-stage knowledge-distillation from a larger to a smaller network for supervised classification is too narrow. For this architectural modification to be properly validated, I would suggest extending to online self-distillation, self-supervised and semi-supervised settings, continual learning, vision transformers and other tasks like object detection. It would be particularly interesting to combine with methods like BYOL, DINO, iBOT, Dense CL etc., which use self-distillation.”
>
> **Our responses** are: **(1)** In the original manuscript, we already evaluated our method with 16 teacher-student network pairs (13/3 pairs on CIFAR-100/ImageNet dataset). In the KD community, a fact is that a large number of mainstream KD methods (including 19 competitors we compared) are proposed for image classification tasks; **(2)** Following your constructive suggestions, we exert all our efforts to improve the experimental evaluation of NORM from the following three aspects:
>
> **Comparison of object detection results on MS COCO dataset.** In the experiments, we also test our method with both same type and different type teacher-student network pairs for object detection. Specifically, we test our method using RetinaNet-ResNeXt101 as teacher detector and RetinaNet-ResNet50 as student detector first, and then using Cascade Mask RCNN-ResNeXt101 as teacher detector and Faster RCNN-ResNet50 as student detector, following the training and test settings used in very recent papers of FGD [2] (CVPR 2022) and MGD [4] (ECCV 2022). We add NORM ($N=4$ for training efficiency) after each output of the FPN neck of RetinaNet-ResNet50/Faster RCNN-ResNet50. Detailed results are summarized in the below two Tables. It can be seen that our method achieves new state-of-the-art mAP results on these two teacher-student detector pairs, validating its effectiveness in handling object detection task.
>
> |Teacher|Student|Distillaton Methods|mAP (%)|AP_S (%)|AP_M (%)|AP_L (%)|
> |--|:--:|:--:|:--:|:--:|:--:|:--:|
> |RetinaNet-ResNeXt101 (41.0)|RetinaNet-ResNet50 (37.4)|FKD [1]|39.6|22.7|43.3|52.5|
> |RetinaNet-ResNeXt101 (41.0)|RetinaNet-ResNet50 (37.4)|CWD [2]|40.8|22.7|44.5|55.3|
> |RetinaNet-ResNeXt101 (41.0)|RetinaNet-ResNet50 (37.4)|FGD [3]|40.7|22.9|45.0|54.7|
> |RetinaNet-ResNeXt101 (41.0)|RetinaNet-ResNet50 (37.4)| MGD [4]|41.0|**23.4**|45.3|55.7|
> |RetinaNet-ResNeXt101 (41.0)|RetinaNet-ResNet50 (37.4)|NORM (ours)|**41.1**|23.3|**45.3**|**55.7**|
>
> |Teacher|Student| Distillaton Methods|mAP (%)|AP_S (%)|AP_M (%)|AP_L (%)|
> |--|:--:|:--:|:--:|:--:|:--:|:--:|
> Cascade Mask RCNN-ResNeXt101 (47.3)|Faster RCNN-ResNet50 (38.4)|FKD [1]|41.5|23.5 |45.0|55.3|
> Cascade Mask RCNN-ResNeXt101 (47.3)|Faster RCNN-ResNet50 (38.4)|CWD [2]|41.7|23.3|45.5|55.5|
> Cascade Mask RCNN-ResNeXt101 (47.3)|Faster RCNN-ResNet50 (38.4)|FGD [3]|42.0|23.8|46.4|55.5|
> Cascade Mask RCNN-ResNeXt101 (47.3)|Faster RCNN-ResNet50 (38.4)|MGD [4]|42.1|23.7|46.4|56.1|
> Cascade Mask RCNN-ResNeXt101 (47.3)|Faster RCNN-ResNet50 (38.4)|NORM (ours)|**42.4**|**24.0**|**46.6**|**56.3**|
>
> [1] Linfeng Zhang and Kaisheng Ma, “Improve object detection with feature-based knowledge distillation: Towards accurate and efficient detectors”, ICLR 2021.
>
> [2] Changyong Shu, et al., “Channel-wise knowledge distillation for dense prediction”, ICCV 2021.
>
> [3] Zhendong Yang, et al., “Focal and global knowledge distillation for detectors”, CVPR 2022.
>
> [4] Zhendong Yang, et al., "Masked Generative Distillation", ECCV 2022.
>
> **Comparison of semantic segmentation results on Cityscapes dataset.** In the experiments, we test our method with both same type and different type teacher-student network pairs for semantic segmentation. Specifically, we test our method using DeepLabV3-ResNet101 as teacher network and DeepLabV3-ResNet18 as student network first, and then using PSPNet-ResNet101 as teacher network and DeepLabV3-ResNet18 as student network, following the training and test settings used in very recent papers of CIRKD [4] (CVPR 2022) and MGD [5] (ECCV 2022). We add NORM ($N=4$ for training efficiency) after the last feature layer of DeepLabV3-ResNet18. Detailed results are summarized in the below two Tables. It can be seen that our method achieves new state-of-the-art results on these two teacher-student network pairs, validating its effectiveness in handling semantic segmentation task.
>
> |Teacher |Student|Distillaton Methods|Performance (mIOU, %)|
> |--|:--:|:--:|:--:|
> |DeepLabV3-ResNet101 (78.07)|DeepLabV3-ResNet18 (74.21)| SKD [1]|75.42|
> |DeepLabV3-ResNet101 (78.07)|DeepLabV3-ResNet18 (74.21)|IFVD [2]|75.59|
> |DeepLabV3-ResNet101 (78.07)|DeepLabV3-ResNet18 (74.21)|CWD [3]|75.55|
> |DeepLabV3-ResNet101 (78.07)|DeepLabV3-ResNet18 (74.21)|CIRKD [4]|76.38|
> |DeepLabV3-ResNet101 (78.07)|DeepLabV3-ResNet18 (74.21)|MGD [5]|n/a|
> |DeepLabV3-ResNet101 (78.07)|DeepLabV3-ResNet18 (74.21)|NORM (ours)|**77.03**|

---

> > ### Comment · Reviewer_xShr · 2022-11-23
> > **Feedback on responses part 5**
> >
> > I acknowledge that the new experiments are a valuable addition and successfully extend the scope of the proposed method in one of the suggested directions.

---

> > > ### Author Response · Authors · 2022-11-23
> > > **Responses to "Feedback on responses part 5"**
> > >
> > > Thank you so much for the recognition of our responses.

---

> ### Author Response · Authors · 2022-11-18
> **Responses to Official Review by Reviewer xShr: Part 4**
>
> 5.**To your comments** “In Table 1, NORM+KD beats the competition only in two columns where many or all of the competitors are missing. In Table 2, it is best in more columns but again many of the competitors are missing. I do not find these results conclusive.
>
> **Our responses** are: **(1)** These results in Table 1 and Table 2 of the original manuscript are conclusive. Note that we test our method with all 13 teacher-student network pairs reported in the paper of CRD (Tian et al., ICLR 2020) for comprehensive experiments. However, subsequent top-performing methods including SRRL (Yang et al., ICLR 2021), SemCKD (Chen et al., AAAI 2021), ReviewKD (Chen et al., CVPR 2021), SimKD (Chen et al., CVPR 2022) and DistPro (Deng et al., ECCV 2022) just report their performance for some but not all of these 13 teacher-student network pairs on CIFAR-100 dataset. That is, in Table 1 and Table 2, the columns with “n/a” for the missing results of some competitors are just due to the above fact, but are not missed by us; **(2)** A similar fact also exists to Table 3 and Table 6 for the performance comparison on ImageNet dataset; **(3)** Actually, we provide so far most comprehensive performance comparisons (including 19 competitors) on both CIFAR-100 and ImageNet datasets. Particularly, our method shows clear margins (see Table 3 and Table 6) for all 3 teacher-student network pairs on large-scale ImageNet dataset.
>
>
> 6.**To your comments regarding an improved ablation of designs for many-to-one representation matching** “An ablation of a "reverse" idea in addition to "Contracting teacher representation" would be to match 1 representation of the student to N representations of the teacher. N-to-N could also be explored.”
>
> **Our responses** are: **(1)** Thanks for your constructive suggestions. Accordingly, we perform a new set of ablative experiments on ImageNet dataset with ResNet34 as teacher and ResNet18 as student. In the experiments, we consider 5 many-to-one representation matching designs: (a) our default design, (b) a reversed design (contracting the teacher representation by a single FT to have $1/N$ times feature channels than the student’s, without using our linear student FT module), (c) a reversed design (using $N$ independent FTs to the teacher representation, without using our linear student FT module), (d) a paired design (using $N$ independent FTs to the teacher representation and another $N$ independent FTs to the student representation), and (e) a paired design (using $N$ independent FTs to the teacher representation, and using our linear student FT module). Detailed results are summarized in the below Table. It can be seen that all five designs improve the performance of the student network, validating the importance of the many-to-one representation matching concept. Comparatively, our default design achieves the best performance, and the designs that apply a single FT or multiple FTs to the teacher representation show obviously worse performance due to the information loss (that is, it is important to preserve the intact pre-trained teacher representation); **(2)** We also provide a deep discussion and more ablative experiments on where the effectiveness of our many-to-one representation matching method comes from. You are referred to **our responses to the first question of Reviewer Xp75**.
>
> |Methods|Top-1 accuracy(%)|
> |--|:--:|
> |Baseline|70.13|
> |NORM (default, student-to-teacher matching routes $N:1$) |**72.14**|
> |Reversed NORM (1 FT to contract teacher features, w/o our linear student FT module, $N:1$) |70.82|
> |Reversed NORM ($N$ teacher FTs, w/o our linear student FT module, $1:N$)|71.18|
> |Paired NORM ($N$ teacher FTs, $N$ student FTs, $N:N$)|70.62|
> |Paired NORM ($N$ teacher FTs, w/ our linear student FT module, $N:N$)|71.78|
>
> 7.**To your comments** “Appendix A.1 appears to be an overkill. The composition of linear layers is obviously linear.”
>
> **Our responses** are: **(1)** Yes, it is well known that the composition of linear layers is also linear; **(2)** However, we add Appendix A.1 for two purposes. **Firstly**, our linear feature transform (FT) module has a residual connection (identity mapping, as shown in Figure 1), and Appendix A.1 shows that this design also maintains the absorbable property. **Secondly**, modern neural networks typically have a global average pooling (GAP) layer before the fully connected (FC) layer, and Appendix A.1 shows that GAP does not affect the absorbable property of our student FT module at inference. That is, the position of the GAP is not changed when merging our linear FT module into the FC layer, making no architectural modifications to the original student network at inference; **(3)** We believe that Appendix A.1 is necessary to make our paper self-contained. Please permit us to retain it in the manuscript.

---

> > ### Comment · Reviewer_xShr · 2022-11-23
> > **Feedback on responses part 4**
> >
> > I thank the authors for the very detailed responses. My feedback:
> >
> > 1. **Comparisons**. This partially addresses my concern. I did not mean that it is the authors' fault to have missing numbers in the tables. However, because it is common that different authors may report on different settings, a common way to address this is to choose one or two best performing competitors that provide code, reproduce them, and then complete the tables accordingly. This would provide more solid comparisons.
> >
> > 2. **Ablations**. These new experiments are very well appreciated and add much value to the paper.
> >
> > 3. **Proof**. Of course the authors are free to choose, but my advice is to at least add a summary of the proof at the beginning. This would ease a lot the reader. I went through the proof in detail and indeed there is no surprise. In fact, even a formal proof wouldn't need more than a paragraph.

---

> > > ### Author Response · Authors · 2022-11-23
> > > **Responses to "Feedback on responses part 4"**
> > >
> > > Thank you so much for the recognition of our responses, and kind suggestions.
> > >
> > > We will continue to improve experimental comparisons and add a summary of the proof at the beginning, so as to further improve our paper during the final paper revision.

---

> ### Author Response · Authors · 2022-11-18
> **Responses to Official Review by Reviewer xShr: Part 3**
>
> 4.**To your comments regarding a discussion of our method and the recent papers (e.g., BYOL, DINO, Dense CL and iBOT) in self-supervised learning research with self-distillation settings** “Another very related idea is to match N features of the student to N features of the teacher by referring to dense features of different image patches. This has been explored in self-distillation settings for self-supervision, e.g. (Dense CL) Wang et al. 2021. Dense Contrastive Learning for Self-Supervised Visual Pre-Training. (iBOT) Zhou et al. 2022. iBOT: Image BERT Pre-Training with Online Tokenizer. Such dense matching is an alternative way "to introduce many parallel knowledge transfer routes." A discussion is in order. A comparison would be a great addition.”
>
> **Our responses** are: **(1)** Thanks for mentioning these recent seminal works in self-supervised learning research with self-distillation settings. Indeed, they also have the concept of many parallel matching routes with their corresponding components, **but our method differs with them in focus, formulation and application**. **Firstly**, existing methods of self-supervised learning with self-distillation settings (a particular type of unsupervised learning) aim to learn a proper visual representation of a single backbone (called "encoder", the teacher and student share this encoder) from a large scale data set of unlabeled images, while our work aims to improve the visual representation of a small student network by a pre-trained larger teacher network under condition that both of them are trained on a target set of labelled images; **Secondly**, in formulation, many existing self-supervised learning methods (e.g., SimCL (Chen et al., ICML 2020), MoCo (He et al,, CVPR 2020), SwAV (Caron  et al., NeurIPS 2020), BYOL (Grill et al., NeurIPS 2020), DenseCL  (Wang et al., CVPR 2021) and DINO (Caron et al., ICCV 2021)) define a contrastive predication task that encourages the encoder (backbone) to attract similar (positive) sample views and dispel different (negative) sample views with their corresponding contrastive losses leveraging data augmentation, and some others (e.g., iGPT (Chen et al, ICML 2020), BeiT (Bao et al., ICLR 2022) and MAE (He et al., CVPR 2022)) define a masked reconstruction task that uses a decoder to predict the original image (pixel-wise/patch-wise) given the representation learnt by the encoder (backbone) with their corresponding reconstruction losses leveraging masked image modeling. iBOT (Zhou et al., ICLR 2022) further combines masked image modeling into self-supervised contrastive learning for improved performance. To these methods such as DenseCL and iBOT, self-distillation is performed between the target model (student) and the online model (teacher, an exponential moving average of student parameters) via contrasting dense sample view pairs of the whole image or the patch (some patches may be masked), and both models share the same encoder (backbone). In contrast, our method leverages a linear FT module added after the last convolutional layer of a small student network to formulate many-to-one representation matching scheme between a single teacher-student layer via student feature expansion and splitting, and dense mimicking. All segments of our expanded student features are simultaneously forced to be similar to the teacher features, and there is no contrastive matching and no paired data augmentation to the input image; **Thirdly**, in application, the trained encoder (backbone) of self-supervised methods is typically used to downstream tasks, while the efficient student network trained by our method is for direct deployment. Besides, iBOT, iGPT, BeiT and MAE with masked image modeling are particularly used to train vision transformers under unsupervised learning regime, while our method with pre-trained large teacher models is mainly used to train smaller convolutional neural networks under supervised learning regime; **(2)** Because of the above differences, it is non-trivial to directly combine our method with BYOL, DINO, Dense CL, iBOT and etc. In existing papers, mainstream KD methods do not compare their performance with these self-supervised representation learning methods. Similarly, in the community of self-supervised representation learning, all aforementioned methods also do not compare their performance with mainstream KD methods; **(3)** Actually, CRD (Tian et al., ICLR 2020) extends contrastive learning into KD research by presenting a novel contrastive distillation loss, and in the original manuscript we already validated the effectiveness of combining our method with CRD (see Table 1 and Table 2); **(4)** Furthermore, we perform a new set of experiments on ImageNet dataset to test the effectiveness of our method for training a vision transformer under masked generative learning. **Please see our responses to your comments regarding experiment improvements**.

---

> > ### Comment · Reviewer_xShr · 2022-11-23
> > **Feedback on responses part 3**
> >
> > I thank the authors for the very detailed discussion. I clearly understand the differences in focus and application and I trust the authors will update the paper accordingly.
> >
> > My point "the experimental setting is too narrow" in my initial review meant for example that, since the main idea is on the architecture (expand on non-overlapping segments, then contract again), there is no fundamental reason why to focus on two-stage distillation between two models and not also apply e.g. to online self-distillation of a single model. In particular, I do not see why it is non-trivial to apply NORM to BYOL or DINO for example, those two being maybe the simplest models in the family. If this works, it will greatly widen the applicability of NORM. I strongly suggest to attempt such an experiment even for the camera-ready if the paper is accepted.
> >
> > I understand there are two sub-communities here progressing somewhat independently, but this is clearly not the best strategy as a whole.

---

> > > ### Author Response · Authors · 2022-11-23
> > > **Responses to "Feedback on responses part 3"**
> > >
> > > Thank you so much for the recognition of our responses.
> > >
> > > The discussion of these two different sub-communities was already added in the revised manuscript.
> > >
> > > We would be happy to consider your suggestion on making an attempt to explore the suggested experimental setting, and add the results in the paper during the final paper revision.

---

> ### Author Response · Authors · 2022-11-18
> **Responses to Official Review by Reviewer xShr: Part 2**
>
> 3.**To your comments regarding a discussion of our method and a related work (Littwin and Wolf, CVPR 2016)** “The novelty appears to be ok as I am not aware of such trick for knowledge distillation. However, a similar trick exists of introducing a number of classifiers by Littwin and Wolf 2015, "The Multiverse Loss for Robust Transfer Learning." The differences are that we have N classifiers rather than N subtensors, C classes rather than C channels, cross-entropy rather than distillation loss and an additional constraint that the classifiers are orthogonal. But structurally, the ideas are very similar. Some discussion is in order.”
>
> **Our responses** are: **(1)** Thanks for mentioning this work (Littwin and Wolf, CVPR 2016). Indeed, both this work and our work have the concept of $N$/many with their corresponding components, **but they are different in focus, formulation and application**. **Firstly**, Littwin & Wolf’s work focuses on knowledge transfer of the same network from the source domain (/dataset) to the target domain, while our work focuses on knowledge transfer from a large teacher network to a smaller student network (both of them are trained on the same dataset). Under this context, for Littwin & Wolf’s work, knowledge transfer needs to be considered between different datasets given a single network; and for our work, knowledge transfer needs to be considered between two networks of different architectures trained on the same dataset. Furthermore, unlike our work, there is no problem of cross-network structural or feature dimension differences to Littwin & Wolf’s work; **Secondly**, in formulation, Littwin & Wolf’s work uses multiple sets of classifiers appended on top of the backbone of a given network and trained on the source domain, and finetunes them on the target domain. In the optimization, each set of classifiers is trained using a separate cross entropy loss and an orthogonal constraint for each class probability scores of any two classifiers in the same set. In contrast, our method leverages a linear FT module added after the last convolutional layer of the student network to formulate many-to-one representation matching scheme between a single teacher-student layer via feature expansion, splitting and mimicking (minimizing an $l_2$-norm distance function). Note that, in Littwin & Wolf’s work, any two classifiers in each set are forced to have the fewest correlation, while all segments of our expanded student features are simultaneously forced to be similar to the teacher features; **Thirdly**, in application, Littwin & Wolf’s work uses the ensemble of multiple sets of finetuned classifiers to get improved performance on the target dataset. Our work benefits from the absorbable property of the linear student FT module, introducing no extra parameters or architectural modifications to the efficient student network at inference; **(2)** Actually, there exists a series of KD methods such as ONE (Lan et al., “Knowledge Distillation by On-the-Fly Native Ensemble”, NeurIPS 2018) and PCL (Wu et al., “Peer Collaborative Learning for Online Knowledge Distillation”, AAAI 2021) which also append multiple head branches (i.e., classifiers) on top of the backbone of a target network and use the on-the-fly learnt ensemble or each head branch as the teacher to guide the training of individual head branches. There are more closely related to our work, and in the original manuscript (see the paragraph titled “One-stage KD methods” in Section 2 and Table 3) we already had the discussion and performance comparison of them and our work.

---

> > ### Comment · Reviewer_xShr · 2022-11-22
> > **Feedback on responses part 2**
> >
> > I thank the authors for the discussion. I suggest to add a summary of this discussion in the paper. I well understand the differences of this work to Littwin and Wolf, CVPR 2016, as my initial comments suggest. Indeed, ONE (Lan et al.) is much closer to this work. Out this discussion, a valid baseline arises, which I am not sure if it has been explored. That is, define N non-overlapping segments for the student, each of dimension equal to the number of classes, then for each segment use softmax (also for the teacher) and cross-entropy with the soft targets predicted by the teacher. This would be similar to ONE but for two-stage distillation between two different teacher-student models.

---

> > > ### Author Response · Authors · 2022-11-23
> > > **Responses to "Feedback on responses part 2"**
> > >
> > > Thank you so much for further feedback.
> > >
> > > **Our responses are**: **(1)** We would be more than happy to add a summary of this discussion (of NORM,  the work by Littwin and Wolf, and ONE by Lan et al.) in the paper during the final paper revision (the paper revision is disabled currently), following your kind suggestion; **(2)**  To your mentioned new baseline, actually we had the corresponding experiments on hand, showing that it performs on par with ONE. **Note that** directly applying the softmax function to each of $N$ student feature segments (each of channel dimension equals to the number of classes) is not applicable, as **each of spatial feature dimension is not 1**. That is, a global average pooling (GAP) layer is necessary, then followed with the softmax function softened with a temperature of $T$ (we set $T=4$, which is the same as that for KD in Table 1, Table 2 and Table 3). In our experiments on ImageNet dataset with ResNet34 as teacher and ResNet18 as student, $N=8$ and $\alpha=8$ (i.e., the default settings in our manuscript), the student model trained by this baseline method shows $70.58$% top-1 accuracy, and ONE reports $70.55$% top-1 accuracy (see Table 2 in the paper of ONE, and Table 3 in our manuscript). Comparatively, our standard NORM gets a student model with $72.14$% top-accuracy, showing clear margins against them and validating the superiority of our $N$-to-one feature representation matching to the $N$-to-one logits representation matching under two-stage distillation regime. Accordingly, we will add this ablation in the paper during the final paper revision.
> > >
> > > Now, we believe that the above concern should be addressed. Looking forward to hearing your feedback.

---

> ### Author Response · Authors · 2022-11-19
> **Responses to Official Review by Reviewer xShr: Part 1**
>
> Thank you so much for the detailed and constructive comments, and the recognition of the novelty of our proposed method, the writing, the basic evaluation on image classification benchmarks and ablations. Please see our below responses to your concerns and questions one by one.
>
> 1.**To your comments regarding the unclear definition of the “vanilla logits based distillation”** “The paper is well written and the method is mostly clear except for the distillation loss. In particular, the default is Euclidean distance between features (3), referred to as NORM. In the experiments, the "vanilla logits based distillation" is additionally used, referred to as NORM+KD, citing the original distillation paper (Hinton et al. 2015). But the original distillation loss in (Hinton et al. 2015) is cross entropy with soft targets given by the teacher, whereas the Euclidean distance between logits is only discussed as an approximation when the temperature is high. So the "vanilla logits based distillation" is not clear. It should be given formally as a second option below (3). This is very important given that NORM+KD is the best performing option. It cannot remain undefined.”
>
> **Our responses** are: **(1)** The “vanilla logits based distillation” is **just the cross-entropy loss** defined in (Hinton et al. 2015), which matches the logits of the student network to the target logits produced by the pre-trained teacher network. Typically, for each training sample, the logits of either teacher or student network are the class probabilities from the softmax layer softened with a temperature of $T$. Following the paper of “Contrastive Representation Matching” CRD (Tian et al., ICLR 2020), in the original manuscript we denote this “vanilla logits based distillation” as KD when combing NORM with it for improved performance; **(2)** Following your careful advice, in Section 3.2 of the revised manuscript we add **a paragraph titled “Augmented optimization of NORM”** to formally define this loss term and its combination with NORM.
>
> 2.**To your comments regarding a potential misunderstanding due to the names NORM and NORM+KD** “The names NORM and NORM+KD are also not appropriate, because it is implied that NORM does not include distillation; which is not true as (3) is a form of distillation. It is more appropriate to use NORM+A and NORM+B for some A and B.”
>
> **Our responses** are: **(1)** We are really sorry for such a potential misunderstanding due to the names NORM and NORM+KD (which follow the use of CRD and CRD+KD in the paper of CRD (Tian et al., ICLR 2020)); **(2)** Following your careful advice, in Section 4 of the revised manuscript we replace NORM+KD and NORM+CRD by NORM+A and NORM+B respectively, and clearly clarify what they refer to.

---

> > ### Comment · Reviewer_xShr · 2022-11-22
> > **Feedback on responses part 1**
> >
> > I thank the authors for their clarifications and updates. My feedback:
> >
> > 1. **Logits**. There is still a confusion of terminology here. By reading Hinton et al. 2015 more carefully, by "logits" one refers to "the inputs of the final softmax". This is clear e.g. in eq (1), where $z_i$ are the logits and they are the inputs of the softmax, while the outputs of the softmax are $q_i$, the "class probabilities" (predicted by the student). Similarly in eq (2), $v_i$ are the logits of the teacher, and the corresponding softmax outputs are $p_i$, the "soft target probabilities". From eq (2), it is clear that cross entropy is applied between probabilities $p_i$, $q_i$, not between logits. In the high temperature limit, it is shown that minimizing the cross entropy is equivalent to minimizing the Euclidean distance between logits, but this is only a limit. It does not hold in general. Now, the authors of this work use "logits" to refer to class probabilities, which is confusing. My advice is to use a formal definition of everything, not just the default eq (3). Then, a question will arise: why not also use N non-overlapping segments in the case of NORM+KD and NORM+CL? (Or is this the case already?)
> >
> > 2. **NORM variants**. In my initial comments, I suggested to replace "NORM and NORM+KD" by "NORM+A and NORM+B for some A and B". What the authors did was they replaced "NORM+KD and NORM+CL" by "NORM+A and NORM+B". This is even more confusing. For one thing, I did not mean "NORM+A and NORM+B" literally, but **for some** A and B. My point was that NORM itself is a form of distillation because it brings teacher and student responses close to each other in the feature space, so the "NORM+KD" variant sounds strange. But if they cannot find more appropriate names for the three variants "NORM, NORM+KD and NORM+CL", better leave the original ones.

---

> > > ### Author Response · Authors · 2022-11-23
> > > **Responses to “Feedback on responses part 1”**
> > >
> > > Thank you so much for further feedback.
> > >
> > > **1. Our responses to your feedback on Logits** include two parts.
> > >
> > > **Responses of Part1:** **(1)** Since $L_{logits}$ in Eq. 5 of our revised manuscript is defined as the cross-entropy loss which matches the logits of the student network to the target logits produced by the pre-trained teacher network, it is our original intention to use **the normalized logits** $p_i$/$q_i$, i.e., the class probabilities from the final softmax function of the student/teacher network softened with a temperature of $T$ (we set $T=4$ in all our experiments, as stated in the caption of Table 1),  aiming to have an easy understanding of $L_{logits}$; **(2)** You are correct, the original logits $z_i$ are the outputs of the penultimate layer of a network, i.e., the inputs of the final softmax function; **(3)** We are really sorry for such a confusion of terminology remained in our revised manuscript. We would be more than happy to formally define the “logits” and the “class probabilities”  (the normalized logits) in the paper during the final paper revision (the paper revision is disabled currently), strictly following your thoughtful comments.
> > >
> > > **Responses of Part 2:** To “a question will arise: why not also use $N$ non-overlapping segments in the case of NORM+KD and NORM+CL? (Or is this the case already?)”, **(1)** In the cases of "NORM+KD" and "NORM+CL" in Table 1 and Table 2, "NORM" is our standard NORM (Eq. 4) which already uses $N$ non-overlapping segments, and "KD/CL" uses the "logits/contrastive" outputs from the penultimate layer pair of the teacher and student networks; **(2)** For fair comparisons, we strictly follow the combination settings used in the seminal work CRD (Tian et al., ICLR 2020), and set our "NORM+KD" and "NORM+CL in the way like "CRD+KD"; **(3)** Applying "KD" or "CL" to $N$ non-overlapping student segments will break the concise design flow and hamper the effectiveness of NORM, introducing $N$ auxiliary classifier branches to the student network and leading to significantly worse performance than our standard NORM, you are referred to **"Responses to Feedback on responses part 2"** for an ablative experiment on hand.
> > >
> > > **2. Our responses to your feedback on the appropriate names of NORM variants:** **(1)** Recall that the core contribution of our work is NORM, an N-to-One Representation Matching mechanism, which is dedicated to enabling dense lossless feature distillation routes between a single layer pair of the teacher and student networks. To highlight this, in the revised manuscript we retain to use “NORM” to denote our standard NORM (defined in Eq. 4), and use “NORM+A” and “NORM+B” to denote two augmented NORM variants (denoted as "NORM+KD" and "NORM+CL" in our original manuscript, respectively) which combine our standard NORM with KD (logits based distillation, Hinton et al., Arxiv 2015) and contrastive loss in CRD (Tian et al, ICLR 2020) respectively, as stated in the caption of Table 1. Comparatively, although naming them in this way may not largely improve the conciseness of notations, but it may not seem even more confusing, to the best of our understanding. As you already commented, NORM itself is a form of feature distillation, we think that using “NORM+A” to denote our standard NORM and using “NORM+B” to denote the augmented NORM combined with KD may be confusing too, even to some A and B; **(2)** Besides leaving the original names "NORM, NORM+KD and NORM+CL" as you suggested, another alternative to name them is: "NORM, NORM(*) and NORM(+)". How do you think about it?
> > >
> > > Now, we believe that the above two remaining concerns should be addressed. Looking forward to hearing your feedback.

---

### Official Review · Reviewer_d7Sn · 2022-10-24

**Confidence:** 4
**Correctness:** 3
**Technical Novelty And Significance:** 2
**Empirical Novelty And Significance:** 3
**Recommendation:** 6

**Clarity, Quality, Novelty And Reproducibility:**

This paper is generally well-written and easy to follow. The methodology part lacks some details for reproduction (e.g., eq. 3, $NC_t$). The authors also promised that the code would be released later.

**Strength And Weaknesses:**

Pros:

+ This work put forward a plug-and-use module that is easy to deploy.

+ The method is experimentally compared with many State-of-the-Art methods on ImageNet and CIFAR-100.

+ The paper is in general well-written.


Cons:

- The method seems to be limited to the convolutional architecture.

- The idea of creating multiple heads with the exact same shape and distill the teacher into each of them simultaneously seems intuitive and I wonder why it is benefitial for improving the distillation performance.

- Experiments are only performed on image classification datasets, will it still work on other tasks such as image segmentation?

- In Equation 3, $d(,)$ is undefined. I guess it is a distance function yet the authors still need to make it clarified.

- It is not clear to me how to find the desired channel dimension $NC_t$ for different teacher-student pairs.

**Summary Of The Paper:**

This work present NORM, a  two-stage feature distillation method. It relies on a linear feature transform module that is inserted after the last convolutional layer of the student network to enable a many-to-one representation matching mechanism conditioned on a single teacher-student layer pair via feature expansion, splitting and group-wise mimicking. Experiments are peformed on CIFAR-100 and ImageNet  to show the effectiveness of the proposed method.

**Summary Of The Review:**

The paper is in general well written. The authors put forward a plug-and-use many-to-one presentation learning module to improve the single teacher-student distillation performance. Experiments under different variants on CIFAR100 and ImageNet are implemented to evaluate the effectiveness of the proposed method. However, some details are missing for reproduction. The method seems also limited to convolutional networks and Image classification tasks.

---

> ### Author Response · Authors · 2022-11-18
> **Responses to Official Review by Reviewer d7Sn: Part 5**
>
> 4.**To your comment** “In Equation 3, d(,) is undefined. I guess it is a distance function yet the authors still need to make it clarified.”
>
> **Our responses** are: **(1)** Yes, in Equation 3, $d(\cdot)$ is an $l_2$-norm distance function. Actually, in the original manuscript, we already described this in the paragraph between Equation 2 and Equation 3; **(2)** To further improve the clarity of Equation 3, in the revised manuscript we replace $d(F_{se}^i,F_t)$ by $||F_{se}^i-F_t||_{2}^2$.
>
> 5.**To your comment** “It is not clear to me how to find the desired channel dimension NCt for different teacher-student pairs”.
>
> **Our responses** are: **(1)** Given a teacher-student pair, the number of feature channels $C_t$ from the last convolutional layer of the teacher network is known and fixed. That is, we do not change $C_t$ so as to preserve the intact information learnt by the teacher network (we already clarified this in the Abstract/Introduction/Method sections of the original manuscript); **(2)** In this context, the desired student channel dimension $NC_t$ for different teacher-student pairs only depends on the selection of $N$. In the original manuscript, we already provided an ablation to study the selection of $N$ and the other hyperparameter $\alpha$ (see **the paragraph titled** “The selection of $N$ and $\alpha$”, Figure 2 and Figure 3); **(3)** As clarified in this ablation, we set $N = 8$ for all 16 teacher-student pairs in our main experiments, without tuning $N$ for each teacher-student pair. Indeed, such a simple setting of $N$ is not optimal, but it already shows consistently promising results.
>
> 6.**To your comment on Reproducibility** “The methodology part lacks some details for reproduction (e.g., eq. 3, NCt).”
>
> **Our responses** are: **(1)** To your concerns about Equation 3 and $NC_t$, please refer to our $4^{th}$ and $5^{th}$ sets of responses described above; **(2)** Besides, other implementation details are also provided in the Appendix (see Section A.2) of the original manuscript; **(3)** **Now, in the Supplementary Material**, we provide a clean version of our source code for main image classification experiments on both CIFAR-100 and ImageNet datasets. Next, we will further improve and clean our newly added code for semantic segmentation and objection detection tasks, which will be also released later. We hope it could help the community to advance KD research.
>
> **Finally**, regarding more experiments and discussions that we have made during the rebuttal phase, you are referred to our top-level comments titled **“The Summary of Our Responses to All Official Reviews”**, our responses to the other reviewers, and the revised manuscript.

---

> > ### Comment · Reviewer_d7Sn · 2022-11-29
> > **Response to the authors**
> >
> > Thank the authors for the very detailed response. Most of my concerns are addressed. Though I still feel the idea is somewhat intuitive, I do appreciate the effort the authors do in the rebuttal period and the supplementary experiments are also valuable additions to the initial paper.  I have read all the review comments and author responses and decided to raise my score.

---

> > > ### Author Response · Authors · 2022-11-29
> > > **Thanks for the Recognition of Our Rebuttal**
> > >
> > > Thank you so much for the recognition of our responses. We are glad to see that you have raised your score.
> > >
> > > We will make more efforts to improve our paper further.
> > >
> > > Many thanks for your constructive comments, time and patience.

---

> ### Author Response · Authors · 2022-11-18
> **Responses to Official Review by Reviewer d7Sn: Part 4**
>
> **Comparison of semantic segmentation results on Cityscapes dataset.** In the experiments, we test our method with both same type and different type teacher-student network pairs for semantic segmentation. Specifically, we test our method using DeepLabV3-ResNet101 as teacher network and DeepLabV3-ResNet18 as student network first, and then using PSPNet-ResNet101 as teacher network and DeepLabV3-ResNet18 as student network, following the training and test settings used in very recent papers of CIRKD [4] (CVPR 2022) and MGD [5] (ECCV 2022). We add NORM ($N=4$ for training efficiency) after the last feature layer of DeepLabV3-ResNet18. Detailed results are summarized in the below two Tables. It can be seen that our method achieves new state-of-the-art results on these two teacher-student network pairs, validating its effectiveness in handling semantic segmentation task.
>
> |Teacher |Student|Distillaton Methods|Performance (mIOU, %)|
> |--|:--:|:--:|:--:|
> |DeepLabV3-ResNet101 (78.07)|DeepLabV3-ResNet18 (74.21)| SKD [1]|75.42|
> |DeepLabV3-ResNet101 (78.07)|DeepLabV3-ResNet18 (74.21)|IFVD [2]|75.59|
> |DeepLabV3-ResNet101 (78.07)|DeepLabV3-ResNet18 (74.21)|CWD [3]|75.55|
> |DeepLabV3-ResNet101 (78.07)|DeepLabV3-ResNet18 (74.21)|CIRKD [4]|76.38|
> |DeepLabV3-ResNet101 (78.07)|DeepLabV3-ResNet18 (74.21)|MGD [5]|n/a|
> |DeepLabV3-ResNet101 (78.07)|DeepLabV3-ResNet18 (74.21)|NORM (ours)|**77.03**|
>
> |Teacher|Student|Distillaton Methods|Performance (mIOU, %)|
> |--|:--:|:--:|:--:|
> |PSPNet-ResNet101 (78.34)|DeepLabV3-ResNet18 (73.20)|SKD [1]|73.87|
> |PSPNet-ResNet101 (78.34)|DeepLabV3-ResNet18 (73.20)|IFVD [2]|n/a|
> |PSPNet-ResNet101 (78.34)|DeepLabV3-ResNet18 (73.20)|CWD [3]|75.93|
> |PSPNet-ResNet101 (78.34)|DeepLabV3-ResNet18 (73.20)|CIRKD [4]|n/a|
> |PSPNet-ResNet101 (78.34)|DeepLabV3-ResNet18 (73.20)|MGD [5]|76.02|
> |PSPNet-ResNet101 (78.34)|DeepLabV3-ResNet18 (73.20)|NORM (ours)|**76.51**|
>
> [1] Yifan Liu, et al., “Structured knowledge distillation for semantic segmentation”, CVPR 2019
>
> [2] Yukang Wang, et al., “Intra-class feature variation distillation for semantic segmentation”, ECCV 2020
>
> [3] Changyong Shu, et al., “Channel-wise knowledge distillation for dense prediction”, ICCV 2021.
>
> [4] Chuanguang Yang, et al., "Cross-image relational knowledge distillation for semantic segmentation", CVPR 2022.
>
> [5] Zhendong Yang, et al., "Masked Generative Distillation", ECCV 2022.
>
> **Comparison of object detection results on MS COCO dataset.** In the experiments, we also test our method with both same type and different type teacher-student network pairs for object detection. Specifically, we test our method using RetinaNet-ResNeXt101 as teacher detector and RetinaNet-ResNet50 as student detector first, and then using Cascade Mask RCNN-ResNeXt101 as teacher detector and Faster RCNN-ResNet50 as student detector, following the training and test settings used in very recent papers of FGD [2] (CVPR 2022) and MGD [4] (ECCV 2022). We add NORM ($N=4$ for training efficiency) after each output of the FPN neck of RetinaNet-ResNet50/Faster RCNN-ResNet50. Detailed results are summarized in the below two Tables. It can be seen that our method achieves new state-of-the-art mAP results on these two teacher-student detector pairs, validating its effectiveness in handling object detection task.
>
> |Teacher|Student|Distillaton Methods|mAP (%)|AP_S (%)|AP_M (%)|AP_L (%)|
> |--|:--:|:--:|:--:|:--:|:--:|:--:|
> |RetinaNet-ResNeXt101 (41.0)|RetinaNet-ResNet50 (37.4)|FKD [1]|39.6|22.7|43.3|52.5|
> |RetinaNet-ResNeXt101 (41.0)|RetinaNet-ResNet50 (37.4)|CWD [2]|40.8|22.7|44.5|55.3|
> |RetinaNet-ResNeXt101 (41.0)|RetinaNet-ResNet50 (37.4)|FGD [3]|40.7|22.9|45.0|54.7|
> |RetinaNet-ResNeXt101 (41.0)|RetinaNet-ResNet50 (37.4)| MGD [4]|41.0|**23.4**|45.3|55.7|
> |RetinaNet-ResNeXt101 (41.0)|RetinaNet-ResNet50 (37.4)|NORM (ours)|**41.1**|23.3|**45.3**|**55.7**|
>
> |Teacher|Student| Distillaton Methods|mAP (%)|AP_S (%)|AP_M (%)|AP_L (%)|
> |--|:--:|:--:|:--:|:--:|:--:|:--:|
> Cascade Mask RCNN-ResNeXt101 (47.3)|Faster RCNN-ResNet50 (38.4)|FKD [1]|41.5|23.5 |45.0|55.3|
> Cascade Mask RCNN-ResNeXt101 (47.3)|Faster RCNN-ResNet50 (38.4)|CWD [2]|41.7|23.3|45.5|55.5|
> Cascade Mask RCNN-ResNeXt101 (47.3)|Faster RCNN-ResNet50 (38.4)|FGD [3]|42.0|23.8|46.4|55.5|
> Cascade Mask RCNN-ResNeXt101 (47.3)|Faster RCNN-ResNet50 (38.4)|MGD [4]|42.1|23.7|46.4|56.1|
> Cascade Mask RCNN-ResNeXt101 (47.3)|Faster RCNN-ResNet50 (38.4)|NORM (ours)|**42.4**|**24.0**|**46.6**|**56.3**|
>
> [1] Linfeng Zhang and Kaisheng Ma, “Improve object detection with feature-based knowledge distillation: Towards accurate and efficient detectors”, ICLR 2021.
>
> [2] Changyong Shu, et al., “Channel-wise knowledge distillation for dense prediction”, ICCV 2021.
>
> [3] Zhendong Yang, et al., “Focal and global knowledge distillation for detectors”, CVPR 2022.
>
> [4] Zhendong Yang, et al., "Masked Generative Distillation", ECCV 2022.

---

> ### Author Response · Authors · 2022-11-18
> **Responses to Official Review by Reviewer d7Sn: Part 3**
>
> **(iii) The role of the designs to construct the many-to-one representation matching.** NORM enables the many-to-one representation matching via expanding the student representation to have $N$ times feature channels than the teacher’s. We can also construct multiple student-to-teacher matching routes in other ways. Accordingly, we perform another set of ablative experiments to study the effect of different designs to construct the many-to-one representation matching, following the insightful suggestions by Reviewer xShr. In the experiments, we compare 5 different designs: (a) our default design, (b) a reversed design (contracting the teacher representation by a single FT to have $1/N$ times feature channels than the student’s, without using our linear student FT module), (c) a reversed design (using $N$ independent FTs to the teacher representation, without using our linear student FT module), (d) a paired design (using $N$ independent FTs to the teacher representation and another $N$ independent FTs to the student representation), and (e) a paired design (using $N$ independent FTs to the teacher representation, and using our linear student FT module). Detailed results are summarized in the below Table. It can be seen that all five designs improve the performance of the student network, validating the importance of the many-to-one representation matching concept. Comparatively, our default design achieves the best performance, and the designs that apply a single FT or multiple FTs to the teacher representation show decreased performance due to the information loss (that is, it is important to preserve the intact pre-trained teacher representation).
>
> |Methods|Top-1 accuracy(%)|
> |--|:--:|
> |Baseline|70.13|
> |NORM (default, student-to-teacher matching routes $N:1$) |**72.14**|
> |Reversed NORM (1 FT to contract teacher features, w/o our linear student FT module, $N:1$) |70.82|
> |Reversed NORM ($N$ teacher FTs, w/o our linear student FT module, $1:N$)|71.18|
> |Paired NORM ($N$ teacher FTs, $N$ student FTs, $N:N$)|70.62|
> |Paired NORM ($N$ teacher FTs, w/ our linear student FT module, $N:N$)|71.78|
>
> **(iv) The role of the ways to split the expanded student representation.** In NORM, for simplicity, we sequentially split the expanded student representation into $N$ feature segments. Our last set of ablative experiments is to study the effect of different feature splitting strategies. In the experiments, we test NORM with 3 different feature splitting strategies: (a) sequential feature splitting, (b) random feature splitting, and (c) importance-based feature splitting is which we first sort feature channels in descending order based on the learnt mean values of channel-wise batch normalization parameters at the fifth epoch, and then perform sequential feature splitting to enable our many-to-one representation matching. Detailed results are summarized in the below Table. It can be seen that all three feature splitting strategies show almost the same performance. This is because that there is no semantic channel-wise alignment between the teacher and the expanded student representations before the many-to-one representation matching.
>
> |Methods| Top-1 accuracy (%)|
> |--|:--:|
> |Baseline|70.13|
> |NORM + sequential feature splitting (default)|72.14|
> |NORM + random feature splitting|72.13|
> |NORM + importance-based feature splitting|**72.15**|
>
> **Based on all ablative experiments described above, we validate the roles of the major components of our method, and provide a deep understanding of what enables the distillation effectiveness of NORM.**
>
>
> 3.**To your comment regarding the experiments on other tasks besides image classification** “Experiments are only performed on image classification datasets, will it still work on other tasks such as image segmentation?”
>
> **Our responses** are: **(1)** Accordingly, we perform four sets of experiments to test the effectiveness of our method both on image segmentation with Cityscapes dataset and object detection with MS COCO dataset; **(2)** Detailed experiments are described as below.

---

> ### Author Response · Authors · 2022-11-18
> **Responses to Official Review by Reviewer d7Sn: Part 2**
>
> **Part 2: A systematic study of NORM**, recall that in the original manuscript, we provide several sets of ablative experiments to show: (1) a large $N$ value is much better than $N=1$, see Figure 2; (2) Given the expanded student representation consisting of $N$ feature segments, the more the feature segments used to match the teacher representation, the larger the accuracy gain, see Figure 4; (3) Inserting a linear FT module into different student networks usually does not bring accuracy improvement under individual training, see Table 1, Table 2 and Table 5. **In order to better understand what enables the distillation effectiveness, here we systematically study the major components of NORM from the following four more aspects:**
>
> **Basic experimental settings:** All following ablative experiments are performed on ImageNet dataset with ResNet34 as teacher and ResNet18 as student, $N=8$ and $\alpha=8$ (i.e., the default settings in the original manuscript).
>
> **(i) The role of the learnable ensemble layer.** In light of the above interpretation of NORM, the second linear layer $W_{sc}$ of our student FT module performs a learnable ensemble of $N$ distillation-augmented student feature views via fully connected channel mixing operations. Accordingly, we conduct a set of ablative experiments to explore the role of this component. In the experiments, we compare NORM with 3 different student FT designs: (a) our default two-layer student FT module, (b) a two-layer student FT module with the fixed ensemble layer (i.e., feature channels from different segments are sequentially averaged), and (c) a three-layer student FT module (i.e., having two same-size linear layers as a learnable ensemble). Detailed results are summarized in the below Table. It can be seen that all three student FT modules bring clear accuracy improvements, validating the importance of the second linear layer for ensembling $N$ augmented student feature views. Comparatively, NORM with the learnable ensemble layer is better than NORM with the fixed ensemble layer, and further improved result is attained when using a more complex learnable ensemble (two-layer).
>
> |Methods|Top-1 accuracy (%)|
> |--|:--:|
> |Baseline|70.13|
> |NORM + a default 2-layer student FT module|72.14|
> |NORM + a 2-layer student FT module w/ the fixed ensemble layer|71.91|
> |NORM + a 3-layer student FT module|**72.21**|
>
> **(ii) The role of the student FT module initialization.** Note that $N$ expanded student feature segments generated by the first linear layer of our student FT module enable the many-to-one representation matching. Next, we perform a set of ablative experiments to study the effect of different weights initialization strategies for the first linear layer, following the insightful suggestions by Reviewer iHQs. In the experiments, we compare NORM with 3 different weights initialization strategies: (a) the default weights initialization we used in Pytorch, (b) initialization with a higher weights variance (8$\times$ of the default), and (c) initialization with the same weights to N segments. Detailed results are summarized in the below Table. It can be seen that all three weights initialization strategies get promising performance, showing the robustness of our method to different initialization strategies. Comparatively, initialization with a higher weights variance tends to bring a bit more gain. What’s more, initialization with the same weights to $N=8$ segments is much better than $N=1$. This benefits from the second linear layer which uses fully connected operations for dynamically mixed feature ensembling. As a results, the gradients are different to the weight groups initialized with the same values, producing different student feature segments (we also confirm this by recording and comparing them on the fly) to match the teacher representation.
>
> |Methods|Top-1 accuracy (%)|
> |--|:--:|
> |Baseline|70.13|
> |NORM (N=1)|71.23|
> |NORM (N=8) + default weights initialization |72.14|
> |NORM (N=8) + initialization with a higher weights variance|**72.22**|
> |NORM (N=8) + initialization with the same weights to N student feature segments|72.08|

---

> ### Author Response · Authors · 2022-11-18
> **Responses to Official Review by Reviewer d7Sn: Part 1**
>
> Thank you so much for the detailed and constructive comments, and the recognition of the novelty of the proposed method, the writing, and the experimental evaluation on image classification benchmarks. Please see our below responses to your concerns and questions one by one.
>
> 1.**To your comment regarding the weakness** “The method seems to be limited to the convolutional architecture.”
>
> **Our responses** are: **(1)** Indeed, in the original manuscript, we evaluate our method with 16 teacher-student network pairs (13/3 pairs on CIFAR-100/ImageNet dataset). The main reason for this is that in the community, mainstream KD methods for image classification, e.g., 19 recent KD methods compared in our work, typically use convolutional architectures for performance evaluation. We follow them for fair and easy comparisons; **(2)** Benefited from the simplicity of our method, we can easily use it to other network architectures. Accordingly, we perform a set of new experiments on ImageNet dataset to test the effectiveness of our method with recent vision transformer architectures. Specifically, we use DeiT-Small as teacher network and DeiT-Tiny as student network, following the settings of VITKD [3] (a very recent work posted on Arxiv in September 2022). For the student network, we add NORM ($N=4$ for training efficiency) to the same layers as VITKD. Detailed results are summarized in the below Table. It can be seen that our method also works well on vision transformer architectures, achieving promising results; **(3)** Here, it should be noted that the standard training of a popular vision transformer model is very time consuming due to significantly larger model size and longer training schedule, compared to that of mainstream convolutional networks (e.g., 300 epochs for DeiTs vs. 100 epochs for ResNets). One run of training the above teacher-student pair of vision transformers by NORM needs about 2.5 days with 8 Nvidia Tesla V100-SXM3 GPUs; **(4)** Besides, we also provide multiple sets of experiments on Cityscapes and MS COCO datasets to show the effectiveness of our method in handling semantic segmentation and object detection tasks. Again, our method attains leading performance. **You are referred to our responses to reviewer Xp75 for detailed comparisons**.
>
> |Teacher|Student|Distillaton Methods|Top-1 accuracy (%)|
> |--|:--:|:--:|:--:|
> |DeiT-Small(80.69)|DeiT-Tiny (74.42)|KD [1]|75.01|
> |DeiT-Small(80.69)|DeiT-Tiny (74.42)|NKD [2]|75.48|
> |DeiT-Small(80.69)|DeiT-Tiny (74.42)|VITKD [3]|75.40|
> |DeiT-Small(80.69)|DeiT-Tiny (74.42)|NORM (ours)|**76.05**|
>
> [1] Geoffrey Hinton, et al, “Distilling the knowledge in a neural network”, arXiv preprint arXiv:1503.02531, 2015.
>
> [2] Zhendong Yang, et al., "Rethinking knowledge distillation via cross-entropy", arXiv preprint arXiv: 2208.10139, 2022.
>
> [3] Zhendong Yang, et al., "ViTKD: Practical Guidelines for ViT feature knowledge distillation", arXiv preprint arXiv:2205.01529, 2022.
>
> 2.**To your comment regarding where the effectiveness of our many-to-one representation matching method comes from** “The idea of creating multiple heads with the exact same shape and distill the teacher into each of them simultaneously seems intuitive and I wonder why it is beneficial for improving the distillation performance.”
>
> **Our responses consist of two Parts**:
>
> **Part 1: Interpretation of NORM**, the formulation of NORM can be interpreted as a way of learning a dynamically mixed feature ensemble over multiple augmented views of the same student representation by forcing them to mimic the intact teacher representation simultaneously. More precisely, in NORM, the first linear layer $W_{se}$ of our student feature transform (FT) module acts as a set of independently initialized feature transforms to generate $N$ channel-expanded views of the student feature $F_s$ whose representation abilities are then parallelly augmented by the distillation supervision from the intact teacher feature $F_t$, and the second linear layer $W_{sc}$ performs a learnable ensemble of $N$ distillation-augmented student feature views via fully connected channel mixing operations (which make the feature ensemble $F_{sc}$ has the same size to $F_s$, guaranteeing the absorbable property of NORM at inference). Our mixed student feature ensemble learning further benefits from the standard cross-entropy loss supervised by the ground truth labels. Accordingly, in Section 3.2 of the revised manuscript we add **a paragraph titled “Interpretation of NORM”**.

---

### Official Review · Reviewer_Xp75 · 2022-11-01

**Confidence:** 4
**Correctness:** 4
**Technical Novelty And Significance:** 3
**Empirical Novelty And Significance:** 3
**Recommendation:** 8

**Clarity, Quality, Novelty And Reproducibility:**

The paper is written with satisfying clarity and quality. I find the idea novel and the results promising, though providing some justification for the effectiveness of the method would further strengthen the work.

On reproducibility, I did not find code submission. But I believe a normal Ph.D. student would have sufficient information to implement the method based on the information provided.

**Strength And Weaknesses:**

Strength:

+ The paper is well written with a clear description of the method and experiments. I also like that the method is simple and seems straightforward to implement based on the provided information.

+ The experiment results seem strong compared with other baseline methods. These methods are up to date.

+ The idea of having N representation matching objectives is interesting. The fact that different transform groups differ only in initialization but together improve accuracy is very intriguing.

Weakness/suggestions:

+ Though effective, there seems to be no obvious motivation to construct multiple transforms at first.  I would like to see some discussion or theoretical analysis on where the effectiveness comes from or what it suggests for the community of model distillation. Because a transform in each group can be considered as an independently initialized tiny model in an ensemble,  could it be possible that it has any connection with ensembling or mixture-of-expert methods? This is more of a question not explored than a weakness, but having this discussion may greatly amplify the impact of this work.

+ Most experiments are conducted on classification tasks. It may be beneficial to show the results in other tasks, such as segmentation and object detection, where distillation is also widely used.

**Summary Of The Paper:**

This paper proposes a method for feature distillation between a teacher model and a student model. It aims to improve the student model's accuracy by utilizing knowledge from the more capable teacher model.

The method adds a linear residual module after the last convolution layer of the student model. The module first transforms the student features to N times the number of channels for the teacher model. Then the expanded student features are grouped into N equal-width groups. Each group is compared with the teacher's features for computing the distillation loss. The distillation loss is optimized in conjunction with the task(classification) loss to learn the student model parameters.

The method is examined on multiple classification benchmarks with different teacher/student pairs. It achieves competitive accuracy with other baseline methods. The authors provide an analysis of the effectiveness of the proposed module through a comparative study.

**Summary Of The Review:**

Overall I feel the submission is of good quality and has satisfying results. My initial recommendation is to accept this paper. A few points as listed in the weakness section describe potential improvements that could be made before publication.

---

> ### Author Response · Authors · 2022-11-18
> **Responses to Official Review by Reviewer Xp75: Part 4**
>
> **Comparison of object detection results on MS COCO dataset.** In the experiments, we also test our method with both same type and different type teacher-student network pairs for object detection. Specifically, we test our method using RetinaNet-ResNeXt101 as teacher detector and RetinaNet-ResNet50 as student detector first, and then using Cascade Mask RCNN-ResNeXt101 as teacher detector and Faster RCNN-ResNet50 as student detector, following the training and test settings used in very recent papers of FGD [2] (CVPR 2022) and MGD [4] (ECCV 2022). We add NORM ($N=4$ for training efficiency) after each output of the FPN neck of RetinaNet-ResNet50/Faster RCNN-ResNet50. Detailed results are summarized in the below two Tables. It can be seen that our method achieves new state-of-the-art mAP results on these two teacher-student detector pairs, validating its effectiveness in handling object detection task.
>
> |Teacher|Student|Distillaton Methods|mAP (%)|AP_S (%)|AP_M (%)|AP_L (%)|
> |--|:--:|:--:|:--:|:--:|:--:|:--:|
> |RetinaNet-ResNeXt101 (41.0)|RetinaNet-ResNet50 (37.4)|FKD [1]|39.6|22.7|43.3|52.5|
> |RetinaNet-ResNeXt101 (41.0)|RetinaNet-ResNet50 (37.4)|CWD [2]|40.8|22.7|44.5|55.3|
> |RetinaNet-ResNeXt101 (41.0)|RetinaNet-ResNet50 (37.4)|FGD [3]|40.7|22.9|45.0|54.7|
> |RetinaNet-ResNeXt101 (41.0)|RetinaNet-ResNet50 (37.4)| MGD [4]|41.0|**23.4**|45.3|55.7|
> |RetinaNet-ResNeXt101 (41.0)|RetinaNet-ResNet50 (37.4)|NORM (ours)|**41.1**|23.3|**45.3**|**55.7**|
>
> |Teacher|Student| Distillaton Methods|mAP (%)|AP_S (%)|AP_M (%)|AP_L (%)|
> |--|:--:|:--:|:--:|:--:|:--:|:--:|
> Cascade Mask RCNN-ResNeXt101 (47.3)|Faster RCNN-ResNet50 (38.4)|FKD [1]|41.5|23.5 |45.0|55.3|
> Cascade Mask RCNN-ResNeXt101 (47.3)|Faster RCNN-ResNet50 (38.4)|CWD [2]|41.7|23.3|45.5|55.5|
> Cascade Mask RCNN-ResNeXt101 (47.3)|Faster RCNN-ResNet50 (38.4)|FGD [3]|42.0|23.8|46.4|55.5|
> Cascade Mask RCNN-ResNeXt101 (47.3)|Faster RCNN-ResNet50 (38.4)|MGD [4]|42.1|23.7|46.4|56.1|
> Cascade Mask RCNN-ResNeXt101 (47.3)|Faster RCNN-ResNet50 (38.4)|NORM (ours)|**42.4**|**24.0**|**46.6**|**56.3**|
>
> [1] Linfeng Zhang and Kaisheng Ma, “Improve object detection with feature-based knowledge distillation: Towards accurate and efficient detectors”, ICLR 2021.
>
> [2] Changyong Shu, et al., “Channel-wise knowledge distillation for dense prediction”, ICCV 2021.
>
> [3] Zhendong Yang, et al., “Focal and global knowledge distillation for detectors”, CVPR 2022.
>
> [4] Zhendong Yang, et al., "Masked Generative Distillation", ECCV 2022.
>
> 3.**To your comment on Reproducibility** “On reproducibility, I did not find code submission. But I believe a normal Ph.D. student would have sufficient information to implement the method based on the information provided.”
>
> **Our responses** are: **(1)** Thanks for recognizing the easy implementation of our method. **Now, in the Supplementary Material**, we provide a clean version of our source code for main image classification experiments on both CIFAR-100 and ImageNet datasets; **(2)** Next, we will further improve and clean our newly added code for semantic segmentation and objection detection tasks, which will be also released later. **(3)** We hope it could help the community to advance KD research.
>
> **Finally**, regarding more experiments and discussions that we have made during the rebuttal phase, you are referred to our top-level comments titled **“The Summary of Our Responses to All Official Reviews”**, our responses to the other reviewers, and the revised manuscript.

---

> > ### Comment · Reviewer_Xp75 · 2022-11-22
> > **Thanks for the detailed feedbacks.**
> >
> > I would like to thank the authors for addressing my questions and providing further studies on interpreting the proposed method.  The experiments on additional vision tasks are also appreciated. I would like to keep my initial rating.
> >
> > Below I provide a perspective on the "interpretation of the NORM" studies. The studies have provided some interesting results and should be included in the paper.
> >
> > Since the "learnable ensemble layer" in the FT module does not have non-linearity before or after, using either one or two linear layer(s) for it should result in the same model due to the associative nature of matrix multiplication. This could explain why the 3-layer student-FT module has very similar results to the normal 2-layer one.
> >
> > Then consider the two-layer FT-module case. The second transform, i.e., the "learnable ensemble layer", can be viewed as performing a transform on each feature segment individually and summing the results together
> >
> > $W_{sc}F_{se}= \sum_{i=1}^N W_{sc}^iF_{se}^i = \sum_{i=1}^N W_{sc}^iW_{se}^iF_{s}^i$,
> >
> > where $W_{sc}^i$ is the $i$-th $C_t \times C_t$ sub-matrix of $W_{sc}$. Thus $W_{sc}^i$ and its counterpart $W_{se}^i$ form a linear sub-network. The FT-module's output without the residual connection, therefore, is the sum of the output of these N linear sub-networks.
> >
> > This indicates that the "fixed ensembling layer" case (a)  might also get similar results because it effectively leads to the same sub-network as the "learnable ensemble layer" case (b) but slightly differs in where to insert the teacher supervision. (a) inserts the supervision after the linear transform while (b) inserts it in-between the linear transform decomposed to two. So I have a little doubt about the effect of "dynamic mixing" and perhaps the necessity of $W_{sc}$.
> >
> > This also suggests that teacher supervision over multiple individually transformed student features (or views) is an effective way of improving feature distillation. Though we still do not know why I believe this discovery already warrants publication.

---

> > > ### Author Response · Authors · 2022-11-23
> > > **Thanks for the Recognition of Our Rebuttal**
> > >
> > > We are so excited that you well recognized our rebuttal. We sincerely thank you for extra comments regarding a truly insightful perspective on our studies for “Interpretation of NORM“. Here, we add some more discussions following your perspective.
> > >
> > > **We fully agree that**, since our student FT module is designed to be fully linear,
> > >
> > > **(1)** Using either one or two  (or more) linear layer (s) as "learnable ensemble layer" in the student FT module theoretically should result in the same model due to the associative nature of matrix multiplication. As you commented, this indeed explains why the 3-layer student FT module has very similar results (slightly better, **72.21% vs. 72.14%** in our ablative experiments) to our default 2-layer one.
> > >
> > > **(2)** Because of the linear property, to the 2-layer student FT module case, the "learnable ensemble layer" (the second linear layer), indeed can be viewed as performing **a learnable transform ($W_{sc}^i$)** on each student feature segment ($W_{se}^i F_s$) individually and **summing** the results together
> > >
> > > $W_{sc}F_{se}= \sum\limits_{i=1}^N W_{sc}^i F_{se}^i = \sum\limits_{i=1}^N W_{sc}^i W_{se}^i F_s$,
> > >
> > > where $W_{sc}^i$ is the i-th $C_s\times C_t$ sub-matrix of $W_{sc}$, $W_{se}^i$ is the i-th $C_t\times C_s$ sub-matrix of $W_{se}$, and $F_s$ is the feature representation from the last convolutional layer of the student network. As you commented, $W_{sc}^i$ and its counterpart $W_{se}^i$ form a linear sub-network. The student FT-module's output without the residual connection, therefore, is the sum of the output of these N linear sub-networks.
> > >
> > > **More discussions based on your comments:** According to (2), this indicates that the "fixed ensembling layer" case might also get similar results (the accuracy gap is relatively small, **71.91% vs. 72.14%** in our ablative experiments) because it effectively leads to the same sub-network **in structure but not in parameters learning** as the "learnable ensemble layer" case, but slightly differs in where to insert the teacher supervision **and the learning property of $W_{sc}^i$**. The "fixed ensembling layer" case inserts the teacher supervision after the linear transform $W_{se}$, **then followed by the fixed $W_{sc}^i$** for each linear sub-network, while the "learnable ensemble layer" case inserts the teacher supervision in-between the linear transform decomposed into two learnable linear layers (**that is, $W_{sc}^i$ is also learnable**). Therefore, the improved accuracy of the "learnable ensemble layer" case to the "fixed ensemble layer" case is mainly due to the effect of learnable $W_{sc}^i$ for "dynamical mixing" of each linear sub-network, i.e., the necessity of a learnable $W_{sc}$. Note that existing works, e.g., ONE (Lan et al., NeurIPS 2018) and PCL (Wu and Gong, AAAI 2021) which were already discussed and compared in our manuscript, have also shown the effectiveness of dynamic (gated) ensemble for online logits based distillation. Besides, the necessity of $W_{sc}$ is also to guarantee the absorbable property of our method at inference.
> > >
> > > These studies added during the rebuttal were already added into the Appendix of the current manuscript. We would be more than happy to include your insightful perspective on our studies for “Interpretation of NORM“ in the paper during the final paper revision (the paper revision is disabled currently).
> > >
> > > Many thanks for your constructive comments, time and patience.

---

> ### Author Response · Authors · 2022-11-18
> **Responses to Official Review by Reviewer Xp75: Part 3**
>
> |Methods| Top-1 accuracy (%)|
> |--|:--:|
> |Baseline|70.13|
> |NORM + sequential feature splitting (default)|72.14|
> |NORM + random feature splitting|72.13|
> |NORM + importance-based feature splitting|**72.15**|
>
> **Based on all ablative experiments described above, we validate the roles of the major components of our method, and provide a deep understanding of what enables the distillation effectiveness of NORM.**
>
> 2.**To your comment regarding the experiments on other tasks besides image classification** “Most experiments are conducted on classification tasks. It may be beneficial to show the results in other tasks, such as segmentation and object detection, where distillation is also widely used.”
>
> **Our responses** are: **(1)** Thanks for your constructive suggestions. Accordingly, we perform four sets of experiments to test the effectiveness of our method both on image segmentation with Cityscapes dataset and object detection with MS COCO dataset; **(2)** Detailed experiments are described as below.
>
> **Comparison of semantic segmentation results on Cityscapes dataset.** In the experiments, we test our method with both same type and different type teacher-student network pairs for semantic segmentation. Specifically, we test our method using DeepLabV3-ResNet101 as teacher network and DeepLabV3-ResNet18 as student network first, and then using PSPNet-ResNet101 as teacher network and DeepLabV3-ResNet18 as student network, following the training and test settings used in very recent papers of CIRKD [4] (CVPR 2022) and MGD [5] (ECCV 2022). We add NORM ($N=4$ for training efficiency) after the last feature layer of DeepLabV3-ResNet18. Detailed results are summarized in the below two Tables. It can be seen that our method achieves new state-of-the-art results on these two teacher-student network pairs, validating its effectiveness in handling semantic segmentation task.
>
> |Teacher |Student|Distillaton Methods|Performance (mIOU, %)|
> |--|:--:|:--:|:--:|
> |DeepLabV3-ResNet101 (78.07)|DeepLabV3-ResNet18 (74.21)| SKD [1]|75.42|
> |DeepLabV3-ResNet101 (78.07)|DeepLabV3-ResNet18 (74.21)|IFVD [2]|75.59|
> |DeepLabV3-ResNet101 (78.07)|DeepLabV3-ResNet18 (74.21)|CWD [3]|75.55|
> |DeepLabV3-ResNet101 (78.07)|DeepLabV3-ResNet18 (74.21)|CIRKD [4]|76.38|
> |DeepLabV3-ResNet101 (78.07)|DeepLabV3-ResNet18 (74.21)|MGD [5]|n/a|
> |DeepLabV3-ResNet101 (78.07)|DeepLabV3-ResNet18 (74.21)|NORM (ours)|**77.03**|
>
> |Teacher|Student|Distillaton Methods|Performance (mIOU, %)|
> |--|:--:|:--:|:--:|
> |PSPNet-ResNet101 (78.34)|DeepLabV3-ResNet18 (73.20)|SKD [1]|73.87|
> |PSPNet-ResNet101 (78.34)|DeepLabV3-ResNet18 (73.20)|IFVD [2]|n/a|
> |PSPNet-ResNet101 (78.34)|DeepLabV3-ResNet18 (73.20)|CWD [3]|75.93|
> |PSPNet-ResNet101 (78.34)|DeepLabV3-ResNet18 (73.20)|CIRKD [4]|n/a|
> |PSPNet-ResNet101 (78.34)|DeepLabV3-ResNet18 (73.20)|MGD [5]|76.02|
> |PSPNet-ResNet101 (78.34)|DeepLabV3-ResNet18 (73.20)|NORM (ours)|**76.51**|
>
> [1] Yifan Liu, et al., “Structured knowledge distillation for semantic segmentation”, CVPR 2019
>
> [2] Yukang Wang, et al., “Intra-class feature variation distillation for semantic segmentation”, ECCV 2020
>
> [3] Changyong Shu, et al., “Channel-wise knowledge distillation for dense prediction”, ICCV 2021.
>
> [4] Chuanguang Yang, et al., "Cross-image relational knowledge distillation for semantic segmentation", CVPR 2022.
>
> [5] Zhendong Yang, et al., "Masked Generative Distillation", ECCV 2022.

---

> ### Author Response · Authors · 2022-11-18
> **Responses to Official Review by Reviewer Xp75: Part 2**
>
> **(ii) The role of the student FT module initialization.** Note that $N$ expanded student feature segments generated by the first linear layer of our student FT module enable the many-to-one representation matching. Next, we perform a set of ablative experiments to study the effect of different weights initialization strategies for the first linear layer, following the insightful suggestions by Reviewer iHQs. In the experiments, we compare NORM with 3 different weights initialization strategies: (a) the default weights initialization we used in Pytorch, (b) initialization with a higher weights variance (8$\times$ of the default), and (c) initialization with the same weights to N segments. Detailed results are summarized in the below Table. It can be seen that all three weights initialization strategies get promising performance, showing the robustness of our method to different initialization strategies. Comparatively, initialization with a higher weights variance tends to bring a bit more gain. What’s more, initialization with the same weights to $N=8$ segments is much better than $N=1$. This benefits from the second linear layer which uses fully connected operations for dynamically mixed feature ensembling. As a results, the gradients are different to the weight groups initialized with the same values, producing different student feature segments (we also confirm this by recording and comparing them on the fly) to match the teacher representation.
>
> |Methods|Top-1 accuracy (%)|
> |--|:--:|
> |Baseline|70.13|
> |NORM (N=1)|71.23|
> |NORM (N=8) + default weights initialization |72.14|
> |NORM (N=8) + initialization with a higher weights variance|**72.22**|
> |NORM (N=8) + initialization with the same weights to N student feature segments|72.08|
>
> **(iii) The role of the designs to construct the many-to-one representation matching.** NORM enables the many-to-one representation matching via expanding the student representation to have $N$ times feature channels than the teacher’s. We can also construct multiple student-to-teacher matching routes in other ways. Accordingly, we perform another set of ablative experiments to study the effect of different designs to construct the many-to-one representation matching, following the insightful suggestions by Reviewer xShr. In the experiments, we compare 5 different designs: (a) our default design, (b) a reversed design (contracting the teacher representation by a single FT to have $1/N$ times feature channels than the student’s, without using our linear student FT module), (c) a reversed design (using $N$ independent FTs to the teacher representation, without using our linear student FT module), (d) a paired design (using $N$ independent FTs to the teacher representation and another $N$ independent FTs to the student representation), and (e) a paired design (using $N$ independent FTs to the teacher representation, and using our linear student FT module). Detailed results are summarized in the below Table. It can be seen that all five designs improve the performance of the student network, validating the importance of the many-to-one representation matching concept. Comparatively, our default design achieves the best performance, and the designs that apply a single FT or multiple FTs to the teacher representation show obviously worse performance due to the information loss (that is, it is important to preserve the intact pre-trained teacher representation).
>
> |Methods|Top-1 accuracy(%)|
> |--|:--:|
> |Baseline|70.13|
> |NORM (default, student-to-teacher matching routes $N:1$) |**72.14**|
> |Reversed NORM (1 FT to contract teacher features, w/o our linear student FT module, $N:1$) |70.82|
> |Reversed NORM ($N$ teacher FTs, w/o our linear student FT module, $1:N$)|71.18|
> |Paired NORM ($N$ teacher FTs, $N$ student FTs, $N:N$)|70.62|
> |Paired NORM ($N$ teacher FTs, w/ our linear student FT module, $N:N$)|71.78|
>
> **(iv) The role of the ways to split the expanded student representation.** In NORM, for simplicity, we sequentially split the expanded student representation into $N$ feature segments. Our last set of ablative experiments is to study the effect of different feature splitting strategies. In the experiments, we test NORM with 3 different feature splitting strategies: (a) sequential feature splitting, (b) random feature splitting, and (c) importance-based feature splitting is which we first sort feature channels in descending order based on the learnt mean values of channel-wise batch normalization parameters at the fifth epoch, and then use sequential feature splitting to enable our many-to-one representation matching. Detailed results are summarized in the below Table. It can be seen that all three feature splitting strategies show almost the same performance. This is because that there is no semantic channel-wise alignment between the teacher and the expanded student representations before the many-to-one representation matching.

---

> ### Author Response · Authors · 2022-11-18
> **Responses to Official Review by Reviewer Xp75: Part 1**
>
> Thank you so much for the thorough and constructive comments, and the recognition of our work including the topic, the novelty of the proposed method, the writing, and the experiments. Please see our below responses to your questions and concerns one by one.
>
> 1.**To your constructive comment regarding a deep discussion on where the effectiveness of our many-to-one representation matching method comes from** “Though effective, there seems to be no obvious motivation to construct multiple transforms at first. I would like to see some discussion or theoretical analysis on where the effectiveness comes from or what it suggests for the community of model distillation. Because a transform in each group can be considered as an independently initialized tiny model in an ensemble, could it be possible that it has any connection with ensembling or mixture-of-expert methods? This is more of a question not explored than a weakness, but having this discussion may greatly amplify the impact of this work.”
>
> **Our responses consist of two Parts**:
>
> **Part 1: Interpretation of NORM**, the formulation of NORM can be interpreted as a way of learning a dynamically mixed feature ensemble over multiple augmented views of the same student representation by forcing them to mimic the intact teacher representation simultaneously. More precisely, in NORM, the first linear layer $W_{se}$ of our student feature transform (FT) module acts as a set of independently initialized feature transforms to generate $N$ channel-expanded views of the student feature $F_s$ whose representation abilities are then parallelly augmented by the distillation supervision from the intact teacher feature $F_t$, and the second linear layer $W_{sc}$ performs a learnable ensemble of $N$ distillation-augmented student feature views via fully connected channel mixing operations (which make the feature ensemble $F_{sc}$ has the same size to $F_s$, guaranteeing the absorbable property of NORM at inference). Our mixed student feature ensemble learning further benefits from the standard cross-entropy loss supervised by the ground truth labels. Accordingly, in Section 3.2 of the revised manuscript we add **a paragraph titled “Interpretation of NORM”**.
>
> **Part 2: A systematic study of NORM**, recall that in the original manuscript, we provide several sets of ablative experiments to show: (1) a large $N$ value is much better than $N=1$, see Figure 2; (2) Given the expanded student representation consisting of $N$ feature segments, the more the feature segments used to match the teacher representation, the larger the accuracy gain, see Figure 4; (3) Inserting a linear FT module into different student networks usually does not bring accuracy improvement under individual training, see Table 1, Table 2 and Table 5. **In order to better understand what enables the distillation effectiveness, here we systematically study the major components of NORM from the following four more aspects:**
>
> **Basic experimental settings:** All following ablative experiments are performed on ImageNet dataset with ResNet34 as teacher and ResNet18 as student, $N=8$ and $\alpha=8$ (i.e., the default settings in the original manuscript).
>
> **(i) The role of the learnable ensemble layer.** In light of the above interpretation of NORM, the second linear layer $W_{sc}$ of our student FT module performs a learnable ensemble of $N$ distillation-augmented student feature views via fully connected channel mixing operations. Accordingly, we conduct a set of ablative experiments to explore the role of this component. In the experiments, we compare NORM with 3 different student FT designs: (a) our default two-layer student FT module, (b) a two-layer student FT module with the fixed ensemble layer (i.e., feature channels from different segments are sequentially averaged), and (c) a three-layer student FT module (i.e., having two same-size linear layers as a learnable ensemble). Detailed results are summarized in the below Table. It can be seen that all three student FT modules bring clear accuracy improvements, validating the importance of the second linear layer for ensembling $N$ augmented student feature views. Comparatively, NORM with the learnable ensemble layer is better than NORM with the fixed ensemble layer, and further improved result is attained when using a more complex learnable ensemble (two-layer).
>
> |Methods|Top-1 accuracy (%)|
> |--|:--:|
> |Baseline|70.13|
> |NORM + a default 2-layer student FT module|72.14|
> |NORM + a 2-layer student FT module w/ the fixed ensemble layer|71.91|
> |NORM + a 3-layer student FT module|**72.21**|

---

### Author Response · Authors · 2022-11-19
**The Summary of Our Responses to All Official Reviews**

Dear Reviewers, Area Chairs and Program Chairs,

We sincerely thank all four reviewers for their thorough and constructive comments. We are glad that the novelty, the writing, the basic experiments and performance of our work have been well recognized by all four reviewers.

In the past two weeks, we carefully improved the experiments (using all computational resources we have), the clarifications and the discussions of our work to address the concerns, the questions and the requests by all four reviewers. **Summarily, we made the following improvements**:

**(1)** To have a better understanding of where the effectiveness of our many-to-one representation matching method comes from, we follow the constructive suggestions by Reviewer Xp75, Reviewer d7Sn and Reviewer xShr, and provide: **(a)** an intrinsic interpretation of our proposed many-to-one representation matching method; **(b)** Based on it, we systematically study the major components of our method through more carefully designed ablative experiments besides the ones reported in the original manuscript, showing more insights by experimental observations.

**(2)** To further improve the experiments, we follow the constructive suggestions/requests from all four reviewers, and provide more extra experiments including **(a)** two sets of experiments on Cityscapes dataset to show that our method can also attain promising performance on semantic segmentation task; **(b)** two sets of experiments on MS COCO dataset to show that our method can also attain promising performance on object detection task; **(c)** a set of experiments on ImageNet dataset to show the effectiveness of our method with a teacher-student pair of recent vision transformer architectures; **(d)** a set of experiments on ImageNet dataset to show the potential of combining our method with a recent masked generative learning strategy for vision transformer based knowledge distillation.

**(3)** We also provide detailed responses to the other concerns/questions/requests raised by each reviewer one by one.

**(4)** In the Supplementary Material, we now provide a clean version of our source code for main image classification experiments on both CIFAR-100 and ImageNet datasets. Next, we will further improve and clean our newly added codes for semantic segmentation and objection detection tasks, which will be also released later. We hope it could help the community to advance KD research.

**Finally**, based on the constructive comments by all four reviewers and our responses, **we carefully revised the manuscript of our work**. We hope our detailed responses and the revised manuscript are helpful to address the concerns, the questions and the requests of all four reviewers.

---

### Decision · Program_Chairs · 2023-01-20

**Decision:**

Accept: poster

**Justification For Why Not Higher Score:**

The proposed method only showed limited gains when generalized to other architectures and tasks (e.g., object detection and semantic segmentation).


**Justification For Why Not Lower Score:**

The proposed modification provides new insights into the commonly-used two-stage distillation paradigm.


**Metareview: Summary, Strengths And Weaknesses:**

This paper was reviewed by four experts in the field. Based on the reviewers' feedback, the decision is to recommend the paper for acceptance to ICLR 2023. All the reviewers acknowledged the value of the proposed plug-and-play modification to the standard two-stage distillation paradigm. The reviewers did raise some valuable concerns that should be addressed in the final camera-ready version of the paper, e.g., adding more details and clarifications for better reproduction. The authors are encouraged to make the necessary changes to the best of their ability. We congratulate the authors on the acceptance of their paper!

**Note From Pc:**

if the above contains the word "oral" or "spotlight" please see: "oral" presentation means -> notable-top-5% and "spotlight" means -> notable-top-25%. As stated in our emails, we are disassociating presentation type from AC recommendations

**Summary Of Ac-Reviewer Meeting:**

N/A